# Plug-and-Play Context Feature Reuse for Efficient Masked Generation

**Xuejie Liu**[1,4], **Anji Liu**[2] , **Guy Van den Broeck**[3], **Yitao Liang**[1]*
[1]Institute for Artificial Intelligence, Peking University
[2]School of Computing, National University of Singapore
[3]Computer Science Department, University of California, Los Angeles
[3]School of Intelligence Science and Technology, Peking University
xjliu@stu.pku.edu.cn, anjiliu@comp.nus.edu.sg
guyvdb@cs.ucla.edu, yitaol@pku.edu.cn

## Abstract

Masked generative models (MGMs) have emerged as a powerful framework for image synthesis, combining parallel decoding with strong bidirectional context modeling. However, generating high-quality samples typically requires many iterative decoding steps, resulting in high inference costs. A straightforward way to speed up generation is by decoding more tokens in each step, thereby reducing the total number of steps. However, when many tokens are decoded simultaneously, the model can only estimate the univariate marginal distributions independently, failing to capture the dependency among them. As a result, reducing the number of steps significantly compromises generation fidelity. In this work, we introduce **ReCAP** (**Re**used **C**ontext-**A**ware **P**rediction), a *plug-and-play* module that accelerates inference in MGMs by constructing low-cost steps via reusing feature embeddings from previously decoded context tokens. ReCAP interleaves standard full evaluations with lightweight steps that cache and reuse context features, substantially reducing computation while preserving the benefits of fine-grained, iterative generation. We demonstrate its effectiveness on top of three representative MGMs (MaskGIT [5], MAGE [27], and MAR [29]), including both discrete and continuous token spaces and covering diverse architectural designs. In particular, on ImageNet256 [7] class-conditional generation, ReCAP achieves up to 2.4× faster inference than the base model with minimal performance drop, and consistently delivers better efficiency–fidelity trade-offs under various generation settings. Our code is publicly available at `https://github.com/liebenxj/ReCAP`.

## 1 Introduction

The remarkable success of sequence modeling in language generation [41, 42] has inspired its adoption in image modeling, where transformer-based models [51, 8, 17] learn the joint distribution over sequences of visual tokens using either autoregressive [11, 10, 48, 22, 44] or non-autoregressive [15, 58, 5, 40] strategies. Among them, Masked Generative Models (MGMs) [27, 29, 54, 52, 56] have emerged as a particularly compelling framework, achieving competitive generation quality while supporting efficient parallel decoding. The advantages in both performance and efficiency have positioned MGMs as a promising alternative to latent diffusion models [45, 39, 2] for high-resolution image generation, with recent work also extending their success to text-to-image synthesis [4, 12].

---

*Corresponding author

39th Conference on Neural Information Processing Systems (NeurIPS 2025).

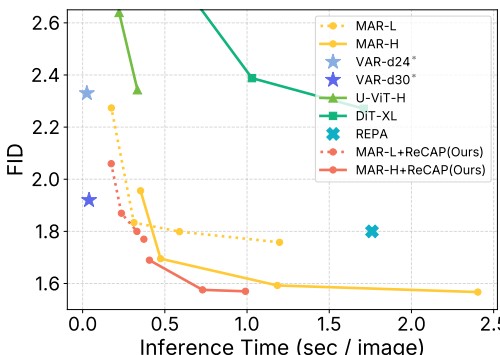

Figure 1: **FID vs. inference time** on ImageNet256 class-conditional generation. As the number of decoding steps increases, MAR [29] achieves better FID but incurs high inference cost. ReCAP significantly accelerates MAR by replacing part of full-eval steps with low-cost steps, achieving 2.4× faster inference for MAR-Huge with minimal quality loss (FID 1.56 vs. 1.57). For fair comparison, we adopt the version of REPA without interval guidance [21], as also reported in the original paper [57]. * denotes the use of KV caching [47] for fast inference.

Despite these promising results, MGMs still face limitations in inference efficiency. As illustrated in Figure 1, applying MAR [29], a state-of-the-art MGM, to class-conditional image generation on ImageNet256 [7] reveals a consistent trade-off: higher sample quality requires more decoding steps, which results in longer inference time. This limitation becomes even more pronounced in large-scale models, where each additional step incurs substantial computational overhead.

We attribute this limitation to a fundamental trade-off in the MGM decoding paradigm. To reduce the total number of generation steps, MGMs decode multiple visual tokens at each step. Ideally, the model should sample from the joint distribution over all tokens being decoded. However, it can only predict the univariate marginal distributions for each token independently due to the sequence-to-sequence nature of Transformers. Therefore, while reducing the number of decoding steps is computationally appealing, it often results in significant degradation in generation quality. To address this dilemma, we pursue an orthogonal direction: *lowering the computational cost per decoding step* while retaining the advantages of fine-grained iterative updates.

Our approach is motivated by a key empirical finding: when only a small number of tokens are updated between decoding steps, the Transformer feature embeddings of the previously decoded context tokens remain largely unchanged. This property is particularly beneficial in settings with many decoding steps, where each step only decodes a small subset of tokens. We leverage this insight to design lightweight decoding steps that reuse the feature representations of context tokens computed during earlier steps. As illustrated in Figure 1, replacing a portion of the full evaluations in the decoding procedure of MAR [29] with these lightweight steps allows us to substantially reduce inference time with minimum reduction on generation quality.

In summary, we propose **ReCAP** (**Re**used **C**ontext-**A**ware **P**rediction), a simple yet effective plug-and-play module for accelerating MGM inference. ReCAP interleaves standard full evaluations with cheaper partial evaluations that reuse cached attention features for the unchanged context tokens. We empirically demonstrate that ReCAP consistently improves quality–efficiency trade-offs across a variety of MGMs, including discrete MGMs (MaskGIT [5], MAGE [27]) and also MGMs with continuous-valued tokens (MAR [29]), covering different architectural designs and evaluation setups. Notably, on ImageNet256 class-conditional generation, ReCAP accelerates inference for MAR-Huge by 2.4×, while preserving its state-of-the-art FID with no architectural edits or additional training.

## 2 Related Work

**Masked Generative Models (MGMs)** originate from non-autoregressive sequence modeling in machine translation [31, 14], offering parallel decoding and faster inference than autoregressive models. Recently, MGMs have been successfully adapted to image generation. As a pioneering work, MaskGIT [5] demonstrated competitive performance on ImageNet using as few as 8 decoding steps, achieving better quality-efficiency trade-offs than diffusion models. Several works aim to improve MGM generation quality: Token-Critic [25] introduces an auxiliary model for guided sampling, MAGE [27] unifies representation and generation via varied masking ratios, and AutoNAT [36] and AdaNAT [37] search for better sampling schedules through optimization-based strategy or reinforcement learning algorithm. However, these methods incur significant cost at longer steps. To bridge the gap with SoTA continuous diffusion models, MAR [29] extends the MGM framework to continuous-valued token spaces, mitigating the information loss from discrete tokenization, and achieves state-of-the-art FID below 2.0 on ImageNet.

**Inference Efficiency in Generative Models.** Accelerating inference is a key challenge in generative modeling. In autoregressive models, techniques such as key-value (KV) caching [51, 47] and speculative decoding [24] reduce redundant computation. In diffusion models, efficiency has improved through advanced solvers [33, 34], classifier-free guidance [19], and guidance interval sampling [21]. While early MGMs benefit from fewer decoding steps and relatively efficient inference [5], achieving state-of-the-art performance remains costly. For example, MAR [29] requires 256 decoding steps to match the quality of leading diffusion models [57, 16], resulting in significant inference overhead.

## 3 Preliminaries of MGMs

In this framework, a raw image $\hat{\mathbf{X}}$ is first encoded into a sequence of $N$ visual tokens $\mathbf{X} = \{X_i\}_{i=1}^N$ using a pretrained tokenizer. Most approaches leverage vector-quantized (VQ) autoencoders [50, 11, 43] as tokenizers, which map image patches to indices in a learned codebook, representing each token $X_i$ as a categorical variable.[2] A transformer-based sequence model is then employed to learn the joint distribution over $\{X_i\}_{i=1}^N$.

To model the sequence data $\mathbf{X}$, MGMs adopt a masked prediction objective, learning to predict masked tokens conditioned on a subset of variables termed the context. This training objective is inspired by masked language modeling tasks used in models like BERT [8, 3, 17] and discrete diffusion models [1]. The model minimizes the following cross-entropy loss:

$$\mathcal{L}(\boldsymbol{x}) = -\mathbb{E}_{r\sim q(\cdot), \boldsymbol{x}_{\mathbf{M}}\sim q_r(\cdot|\boldsymbol{x})} \left[ \sum_{i\in\{i|x_i^{\mathbf{M}}=[\text{MASK}]\}} \log p_\theta(x_i|\boldsymbol{x}_{\mathbf{M}}) \right],$$

where $r$ is a sampled masking ratio and $q_r$ is a masking distribution that replaces an $r$-fraction of tokens in $\boldsymbol{x}$ by [MASK]. At test time, the model begins with an empty context (i.e., all tokens are masked) and gradually decodes a chosen subset of tokens, expanding the context at each step by incorporating the newly decoded tokens. Notably, each step updates multiple tokens in parallel, allowing high-quality generation with significantly fewer decoding steps.

Specifically, the decoding procedure begins with a fully masked sequence $\boldsymbol{x}^{(0)}$. At each step $t \in \{1, \ldots, T\}$, the model predicts token values based on the current sequence $\boldsymbol{x}^{(t-1)}$. We define $n_t$ as the total number of decoded tokens after step $t$, and $\hat{n}_t = n_t - n_{t-1}$ as the number of tokens to be decoded at $t$. Let $\mathcal{M}_t = \{i \mid x_i^{(t-1)} = [\text{MASK}]\}$ denote the masked positions at the beginning of the $t$-th step. The sampler selects a subset $\mathcal{S}_t \subset \mathcal{M}_t$ of size $\hat{n}_t$, either uniformly [29] or proportionally to the model's predicted confidence [5, 27] (see Appendix C). For each $i \in \mathcal{S}_t$, the model samples $x_i^{(t)}$ from the trained MGM $p_\theta(X_i|\boldsymbol{x}^{(t-1)})$.

## 4 The Inference Challenge of MGMs

MGMs offer a powerful framework for high-quality image generation by progressively refining predictions through iterative masked sampling. However, as shown in Figure 1, achieving high-fidelity results typically requires a large number of decoding steps, leading to significant inference costs. For example, MAR-Large [29], a state-of-the-art MGM, achieves an FID of 1.8 using 128 decoding steps, but its performance deteriorates sharply to an FID of 15.9 when constrained to only 8 steps (measured using the official code). For even larger models such as MAR-Huge, inference latency can exceed 2 seconds per image when using hundreds of steps, posing a challenge for the practical deployment of MGMs in latency-sensitive applications.

We attribute this undesirable reliance on numerous decoding steps to the inherent limitations of its parallel decoding procedure. As illustrated in Section 3, in each step $t$, multiple masked tokens are sampled *independently* from the conditional distribution $p(x_i|\boldsymbol{x}^{(t-1)})$, neglecting intricate dependencies among simultaneously unmasked tokens. Similar issues have been identified in discrete diffusion models for masked language modeling [30, 32], where multiple refinement steps are needed to recover coherent outputs due to parallel yet independent updates.

---

[2]Although we describe our method in the context of discrete MGMs, our method is directly applicable to MGMs with continuous-valued tokens (Appendix D).

As a result, reducing the number of decoding steps inherently limits the model's ability to capture inter-token dependencies, thereby degrading generation quality. Using a large number of steps and allowing more incremental and context-aware updates helps alleviate this issue, but at the cost of significant computational overhead. Specifically, unlike causal transformers, which support efficient reuse of intermediate attention states via key-value caching [47], the bidirectional nature of MGMs necessitates recomputing the attention-based features over all $N$ tokens in the sequence at each step, which incurs a $\mathcal{O}(N^2)$ inference cost in each update.

These challenges can be summarized in one central question in our paper: *can we reduce the per-step computation cost while preserving generation quality?* We answer the question in its affirmative by showing that we can cache and reuse feature embeddings of previously decoded tokens with minimum performance drop with the so-called cheap update steps. By balancing the regular and cheap steps, our method achieves a substantially better trade-off between inference speed and generation fidelity.

# 5 Inference Scaling via Context Feature Reuse

To mitigate the inference inefficiency of MGMs, we aim to construct computationally *lightweight* decoding steps that accurately simulate standard many-step MGM decoding at significantly lower cost. Specifically, we interleave the original $T$ full evaluation steps with $T'$ low-cost steps, thereby increasing the number of decoding steps without increasing computational burden proportionally. This allows the model to better capture inter-token dependencies, which leads to better speed–fidelity trade-offs. Importantly, our method requires no change to the model architecture and introduces no additional training cost, making it a simple plug-in mechanism applicable to a broad range of existing MGM frameworks.

In each decoding step, the model has to re-compute Transformer feature embeddings for all tokens in the sequence, even though only a small subset of these tokens are actually modified/unmasked between consecutive steps. This is necessary because the applied bidirectional attention mechanism allows every token's representation to depend on all others; thus, even a small input change can, in principle, propagate globally and alter all token embeddings. However, we hypothesize that when only a few tokens are newly updated, the feature embeddings of the previously decoded context tokens change only slightly, reducing the need for frequent recomputation.

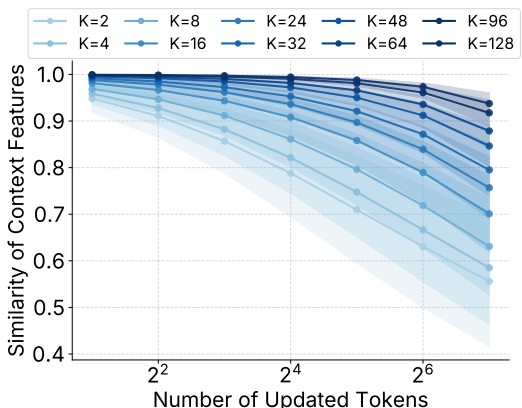

Figure 2: **Context feature stability** during iterative decoding. We measure similarity between context representations before and after token updates, using a pretrained MaskGIT on 50K ImageNet256 samples. At each decoding stage, we extract the input embeddings to the attention module for the $K$ already-decoded tokens. These are average-pooled within each layer to obtain an aggregated context vector. Cosine similarity is computed between these vectors before and after updates and averaged across layers; shaded regions indicate layer-wise standard deviation. Greater stability at larger $K$ supports reusing cached features in later decoding stages.

To validate this hypothesis, we analyze the representations computed by a pretrained MaskGIT [5] model on 50K samples from the ImageNet256 validation set. For each sample, we randomly select $K$ tokens as context and mask the remaining ones, simulating an intermediate decoding state with $K$ already-decoded tokens. As shown in Figure 2, each curve corresponds to a different value of $K$, and the $x$-axis denotes the number of masked tokens subsequently unmasked.

To quantify how the context features evolve during decoding, we incrementally unmask/update a growing number of masked tokens, corresponding to increasing positions along the x-axis, and measure how the feature embeddings of the $K$ given tokens change as a result. Specifically, we extract the input embeddings to the attention module (i.e., pre-QKV projection features), average-pool them across the $K$ context tokens at each layer, and compute the cosine similarity between these aggregated features before and after each

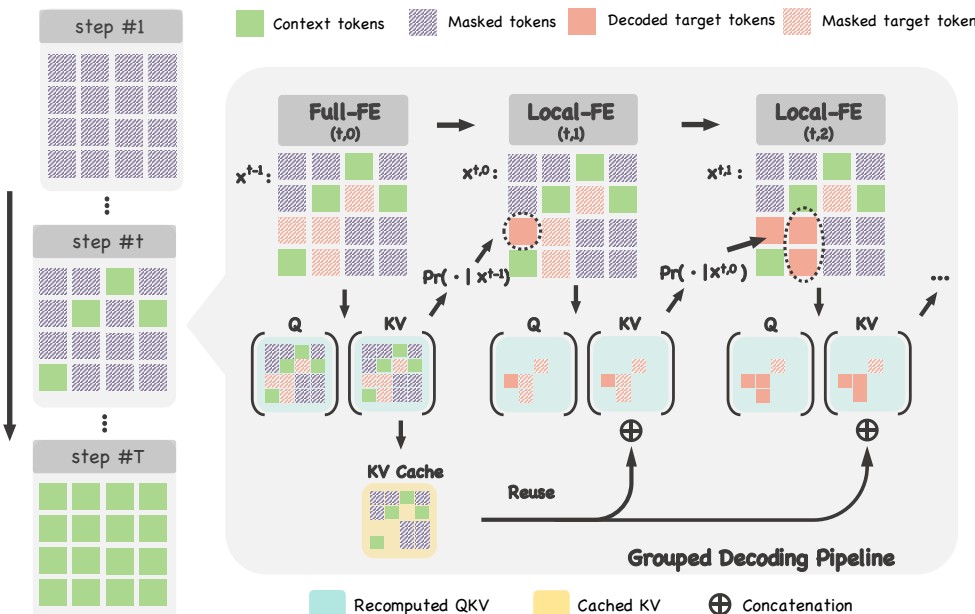

Figure 3: **Grouped Decoding Pipeline with Cached Attention.** Inference is organized into $T$ groups, each performing one *Full-FE* and several *Local-FE* steps. In the Full-FE, full attention is computed over the entire sequence, and KVs for the static context tokens ( ) and other masked tokens ( ) are cached. In each Local-FE, only the QKVs of the target tokens ( ) are recomputed ( ), while the cached KVs ( ) are reused to form the full attention context. The context feature reuse mechanism effectively reduces computation cost in local evaluation steps.

update ($y$-axis). We repeat this process for multiple values of $K \in \{2, 4, 8, \ldots, 128\}$ to simulate decoding states at various stages.

The results in Figure 2 provide strong empirical support for our hypothesis. Across all values of $K$, when only a small number of tokens are updated (left side of the x-axis), the cosine similarity remains close to 1, indicating that the representations of previously decoded tokens change minimally. This suggests that attention features for the decoded context can be safely cached and reused in subsequent steps, enabling more efficient computation.

Moreover, we can observe in Figure 2 that as decoding progresses and the context becomes richer with $K$ increases, the extent of feature drift further diminishes, suggesting that the internal representations associated with context tokens become increasingly stable. This empirical evidence suggests that cached context embeddings can be increasingly reused in later decoding steps, with minimal fidelity loss introduced. Building on these observations, we propose a grouped decoding strategy that interleaves full and partial function evaluations to enable context feature reuse and improve inference efficiency. The overall pipeline is illustrated in Figure 3. Intuitively, the key idea is to cache and reuse the Transformer feature embeddings of unchanged tokens. We implement this by caching and reusing their corresponding key-value (KV) pairs in attention computation, as detailed in the following.

**Grouped Decoding Pipeline.** As illustrated on the left of Figure 3, we organize the MGM inference process into $T$ *grouped decoding stages*, where masked tokens ( ) are progressively converted into decoded context tokens ( ) over time. The detailed structure of each grouped step $t$ is shown in the central gray box of Figure 3. Within each group $t$, a target set of masked tokens $\mathcal{S}_t$ is selected and decoded using multiple light-weight steps. These target tokens are marked in red ( ) and are initially masked ( ). As decoding progresses, they are gradually replaced with decoded tokens ( ).

Each group consists of a **Full Function Evaluation (Full-FE)** at sub-step $(t, 0)$, followed by $l_t$ **Local Function Evaluation (Local-FE)** sub-steps $(t, 1), \ldots, (t, l_t)$. At each sub-step $j \in [0, l_t]$, a disjoint

subset $\mathcal{S}_t^{(j)} \subseteq \mathcal{S}_t$ is decoded and used as context in later sub-steps. At the end of group $t$, all tokens in $\mathcal{S}_t$ will be unmasked, serving as the input context for the next group $t + 1$.

**Full-FE with Cache Construction.** At sub-step $(t, 0)$, i.e., the "Full-FE" panel of Figure 3, we perform a full attention computation over the current sequence $\boldsymbol{x}^{(t-1)}$. This includes computing QKV representations for all tokens. We then cache the KVs for the complement set $\bar{\mathcal{S}}_t$, which consists of the context tokens (█) and unselected masked tokens (▨). These cached KVs (highlighted in █), denoted as $\mathbf{k}_{\bar{\mathcal{S}}_t}^{\text{cached}}, \mathbf{v}_{\bar{\mathcal{S}}_t}^{\text{cached}}$, will be reused in subsequent Local-FEs. Next, we decode the first subset $\mathcal{S}_t^{(0)}$ by sampling from the model distribution $p(X_i \mid \boldsymbol{x}^{(t-1)})$, producing the updated sequence $\boldsymbol{x}^{(t,0)}$.

**Local-FE with Reused KVs.** In each Local-FE sub-step $(t, j)$ for $j \in [1, l_t]$, we decode the subset $\mathcal{S}_t^{(j)}$ based on the current sequence $\boldsymbol{x}^{(t,j-1)}$. Instead of recomputing attention for all tokens, we only recompute the QKVs for the target subset $\mathcal{S}_t^{(j)}$ (highlighted in █). These are then concatenated with the cached KVs to form the full attention context:

$$\mathbf{attn}_{\mathcal{S}_t^{(j)}} = \text{Softmax}\left( \frac{\mathbf{q}_{\mathcal{S}_t^{(j)}} \mathbf{k}_{1:N}^\top}{\sqrt{d_k}} \right) \mathbf{v}_{1:N},$$

where $\mathbf{k}_{1:N} = \text{Concat}(\mathbf{k}_{\mathcal{S}_t^{(j)}}, \mathbf{k}_{\bar{\mathcal{S}}_t}^{\text{cached}})$, $\mathbf{v}_{1:N} = \text{Concat}(\mathbf{v}_{\mathcal{S}_t^{(j)}}, \mathbf{v}_{\bar{\mathcal{S}}_t}^{\text{cached}})$, $d_k$ denotes the dimensionality of the key/value vectors. Each Local-FE then produces $\boldsymbol{x}^{(t,j)}$ by sampling from the model distribution conditioned on $\boldsymbol{x}^{(t,j-1)}$. This process continues until all $l_t$ Local-FEs are completed. While the initial Full-FE incurs a full $\mathcal{O}(N^2)$ cost, each subsequent Local-FE performs a much cheaper update with complexity $\mathcal{O}(\hat{n}_t \cdot N)$, where $\hat{n}_t = |\mathcal{S}_t^{(j)}| \ll N$. Group $t$ then concludes by setting $\boldsymbol{x}^{(t)} := \boldsymbol{x}^{(t,l_t)}$.

**Efficiency and Fidelity Trade-off.** Overall, our method realizes a $(T + T')$-step generation process, where $T' = \sum l_t$ denotes the number of inserted Local-FEs. This strategy effectively adapts the KV caching mechanism to the bidirectional masked generation. Unlike autoregressive transformers—where each decoding step is inherently cache-friendly due to causal masking—bidirectional MGMs require periodic full evaluations to prevent error accumulation. Our grouped decoding framework interleaves exact (Full-FE) and approximate (Local-FE) steps, offering a flexible trade-off between inference speed and generation fidelity.

# 6 Experiment

Our method **ReCAP** (**Re**used **C**ontext-**A**ware **P**rediction) is a plug-and-play approach that can be seamlessly integrated into the inference pipeline of existing MGMs. By interleaving full and partial attention computations, ReCAP significantly reduces the per-step inference cost. In this section, we evaluate whether the use of Local-FEs can effectively lead to efficiency gains and, more importantly, whether it can achieve better trade-offs between generation quality and inference speed.

To this end, we apply ReCAP to three representative MGM baselines and conduct a thorough evaluation. Our experiments span a diverse set of settings, varying in task (class-conditional vs. unconditional generation) and model architecture (decoder-only vs. encoder-decoder). The selected baselines are: i) **MaskGIT** [5]: A widely-used discrete MGM that uses a VQGAN tokenizer [11], followed by a Transformer trained with a BERT-style masked modeling objective, as described in Section 3. ii) **MAR** [29]: A state-of-the-art continuous-valued MGM designed to avoid quantization artifacts by operating on latent embeddings. It reconstructs masked tokens via a per-token diffusion loss. (see Appendix D) iii) **MAGE** [27]: A discrete MGM improving MaskGIT by incorporating a variable mask ratio training objective, which serves as a strong unconditional generation baseline that does not rely on pretrained self-supervised features [28, 38].

Among these, MaskGIT uses an decoder-only architecture, while MAR and MAGE adopt an encoder-decoder architecture following the Masked Autoencoders (MAE) [17]. As demonstrated in the following sections, ReCAP is model-agnostic and can be effectively applied across diverse model designs. We report Fréchet Inception Distance (FID) [18] and Inception Score (IS) [46] following common practice [9]. All inference times are re-evaluated using the official implementations on a single NVIDIA A800 GPU with a default batch size of 200 and reported as time per image.

Table 1: **Performance of MaskGIT w/ and w/o ReCAP** on ImageNet256 class-conditional generation w/o CFG [19]. # Steps = $T + T'$ denotes the total number of decoding iterations. $u$ is the number of initial grouped steps without Local-FEs when applying ReCAP. Results show that ReCAP reliably reduces inference time while maintaining competitive FID.

| # Steps | MaskGIT-r (Full only) | | MaskGIT-r+ReCAP | | | | |
| --- | --- | --- | --- | --- | --- | --- | --- |
| | FID↓ | Time↓ | $u$ | # Full-FE($T$) | # Local-FE($T'$) | FID↓ | Time↓ |
| 16 | 4.46 | 0.095 | 0 | 8 | 8 | 5.02 | 0.055 |
| | | | 8 | 12 | 4 | 4.50 | 0.076 |
| 20 | 4.18 | 0.118 | 10 | 15 | 5 | 4.23 | 0.094 |
| 24 | 4.09 | 0.142 | 10 | 16 | 8 | 4.09 | 0.107 |
| 32 | 3.97 | 0.189 | 12 | 22 | 10 | 3.98 | 0.137 |

## 6.1 Improving MaskGIT with ReCAP

MaskGIT adopts a cosine decoding schedule and a confidence-based token sampler. When adapting ReCAP, we use the same token sampler to obtain $\mathcal{S}_t$ after each Full-FE step (see Appendix B).

Following the decoding schedule used in MaskGIT, early decoding steps reveal only a small number of tokens, while later steps decode progressively more. Recall in Figure 2, we show that context features become more stable as more tokens are decoded. Therefore, we introduce Local-FEs primarily in the later grouped steps. Specifically, for MaskGIT+ReCAP, let $T$ denote the number of grouped decoding steps, which also corresponds to the number of Full-FEs (recall from Figure 3 that each group begins with a Full-FE). We define $u$ as the number of initial steps that only perform Full-FE, i.e., $l_{1:u} = 0$. After step $u$, we insert one Local-FE per step, i.e., $l_{u+1:u+T'} = 1$, where $T'$ is the total number of Local-FEs and the total number of decoding steps equals $T + T'$.

To assess the impact of ReCAP, we conduct a controlled experiment by fixing the total number of decoding steps $T + T'$ and adjusting the allocation between Full- and Local-FEs for ReCAP via the parameter $u$. As a baseline, we replace all Local-FEs with Full-FEs, denoted **MaskGIT-r (Full only)**, which serves as a principal "upper bound" in performance but incurs higher inference cost. Here, MaskGIT-r represents our re-implemented MaskGIT with an enhanced sampling schedule, demonstrating improved inference scaling with increasing decoding steps compared to the original MaskGIT in Figure 4 (see Appendix A). The FIDs and the corresponding runtimes per image of both methods are presented in Table 1. In each row using a certain number of total steps, **MaskGIT-r+ReCAP** achieves notable inference speedups by replacing a subset of Full-FEs with cheaper Local-FEs. For example, with 32 total steps, ReCAP reduces inference time from 0.189s to 0.137s per image while maintaining a

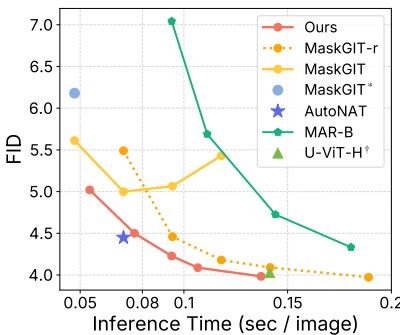

Figure 4: **FID vs. inference time** for MaskGIT variants and comparative models. *: taken from the MaskGIT paper [5]. †: with CFG [19]. U-ViT [2] adopts 7 sampling steps in this figure.

nearly identical FID (3.97 vs. 3.98). With 20 steps, although the FID slightly increases from 4.18 to 4.23, the inference time drops to match that of the 16-step baseline—while significantly outperforming it (FID 4.46). These results demonstrate that ReCAP effectively improves quality-speed trade-offs of the base MaskGIT, achieving comparable or better performance at lower cost. For a more detailed ablation study of the hyperparameters, e.g., , $T$, $T'$, and $l$, please refer to Appendix E.4.

We further visualize this improvement in Figure 4, comparing ReCAP against various strong baselines. The "MaskGIT-r (Full only)" and "MaskGIT-r+ReCAP" in Table 1 correspond to MaskGIT-r and Ours in Figure 4, respectively. By substantially reducing inference time while preserving generation quality, ReCAP enhances the base model's inference-scaling behavior. Moreover, without relying on classifier-free guidance (CFG) [19], our ReCAP-augmented model achieves a more favorable quality–efficiency trade-off compared to advanced continuous diffusion models such as U-ViT-H† [2], which incorporate both DPM solvers [33, 34] and CFG. Our performance is also comparable to AutoNAT [36], which improves sampling via an extensive hyperparameter search. However, AutoNAT does not generalize well to longer decoding schedules. In contrast, ReCAP is broadly applicable as long as the performance of the base model improves as we increase the number of steps.

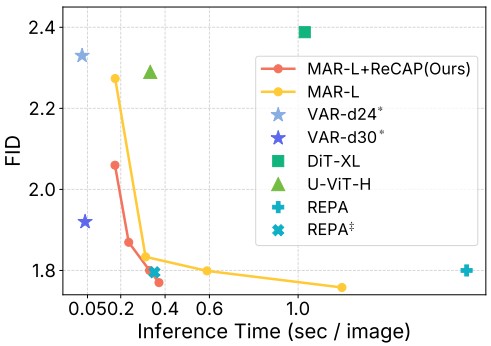 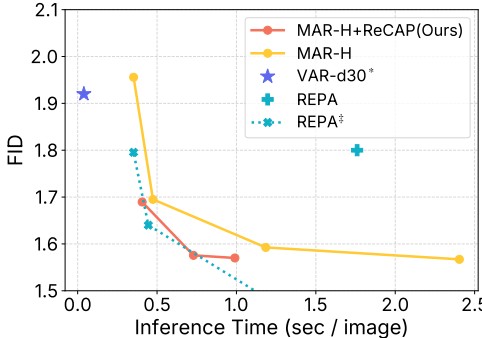

Figure 5: **Speed/Performance trade-off** for MAR variants and SoTA baselines. ReCAP consistently improves inference efficiency of MAR-Large and -Huge. VARs [48] are SoTA AR models performing next-scale prediction, * denotes the use of KV caching [47]. REPA [57], a SoTA flow-matching model relying on vision foundation models [38], ‡ denotes the use of advanced guidance interval sampling [21]. DPM solvers [33, 34] augment DiT [39] and U-ViT [2].

We provide a comprehensive comparison against more strong generative baselines in Appendix E.1, such as Token-Critic [25] and DPC [26] with learnable guidance, StraIT employing hierarchical modeling [40]. Notably, our MaskGIT-r baseline—obtained by simply adjusting the sampling schedule and increasing decoding steps—already outperforms these approaches with more complex architectures or guidance mechanisms. ReCAP further improves MaskGIT-r by offering plug-and-play efficiency gains, requiring no additional training or architectural modifications.

## 6.2    ReCAP for Continuous-Valued MGMs

We further adapt ReCAP to the state-of-the-art continuous-valued MGMs, MAR [29]. Unlike MaskGIT, MAR adopts a MAE-style [17] encoder-decoder architecture, where the encoder operates only on unmasked tokens, while the decoder process the full sequence. Both components employ bidirectional full attention. To fully improve efficiency, we incorporate ReCAP into both the encoder and decoder of MAR-Large and MAR-Huge. For sampling, MAR uses a random sampler for token selection, and additionally requires a denoising MLP process for token reconstruction.[3] Following Section 6.1, we set $l_{1:u} = 0$ and $l_{u+1:u+T'} = 1$ with $u = \frac{T+T'}{2}$ by default. Other sampling configurations, such as the CFG scale, all follow the official MAR codebase.

As shown in Figure 5, MAR-L and MAR-H require many decoding steps to achieve state-of-the-art FID, resulting in considerable inference cost. Augmenting with ReCAP offers substantial speedup—achieve up to 2∼2.4× faster inference while maintaining the performance (±0.01 FID) to their original counterparts. Furthermore, MAR+ReCAP matches the best performance of REPA [57], which is obtained by adopting the advanced guidance interval sampling [21]. Notably, REPA is a leading flow-matching model that leverages self-supervised features from vision foundation models [38] for training. While autoregressive models like VAR [48] remain more efficient due to the use of KV caching [47], MAR+ReCAP outperforms them in generation quality.

We further benchmark our method against a wide range of state-of-the-art generative models under classifier-free guidance, as shown in Figure 5. Our ReCAP-augmented variants demonstrate substantial improvements in inference efficiency. For instance, MAR-H+ReCAP with (96+32) decoding steps, i.e., 96 Full-FEs and 32 Local-FEs, achieves a FID of 1.57—closely matching the original MAR-H at 256 steps (FID 1.56)—while reducing inference time from 2.4s to 1.0s per image. Likewise, MAR-L+ReCAP with (64+20) steps achieves a FID of 1.80 in just 0.33s, outperforming the baseline MAR-L (FID 1.83 at 64 steps) while incurring only an additional 0.02s of inference time from 20 local evaluations. These results highlight the plug-and-play effectiveness of ReCAP in accelerating inference for continuous-valued masked models, enabling strong efficiency–quality trade-offs even when using classifier-free guidance. To further validate ReCAP's generality, we

---

[3]In MAR, inference cost stems from both transformer attention and the per-token diffusion MLP. A detailed cost breakdown is provided in Appendix E.2.

Table 2: **Benchmarking with state-of-the-art models** on ImageNet256 class-conditional generation with classifier-free guidance. We compare ReCAP against representative diffusion baselines such as U-ViT[2], DiT[39], and REPA[57], each evaluated under varying sampling steps to illustrate their step-scaling behavior. Notably, to achieve a FID of 1.8, the original MAR-L requires ∼0.6s per image, whereas our MAR-L+ReCAP only needs 0.33s, outperforming the SoTA REPA‡ (0.35s). ‡ denotes the use of interval guidance [21]

| Method | # Params | NFE | FID↓ | IS↑ | Time (s)↓ |
|---|---|---|---|---|---|
| **Diffusion Models** | | | | | |
| ADM-G [9] | 554M | 250×2 | 4.59 | 186.7 | – |
| VDM++ [20] | 2B | 512×2 | 2.12 | 267.7 | – |
| LDM-4-G [45] | 400M | 250×2 | 3.60 | 247.7 | – |
| U-ViT-H/2 [2] | 501M | 50×2 | 2.29 | 263.9 | 0.33 |
| | | 25×2 | 2.64 | 262.9 | 0.22 |
| | | 7×2 | 4.03 | 234.5 | 0.14 |
| DiT-XL/2 [39] | 675M | 250×2 | 2.27 | 276.2 | 1.71 |
| | | 150×2 | 2.39 | 271.6 | 1.03 |
| | | 50×2 | 3.75 | 243.5 | 0.35 |
| DiffiT [16] | 561M | 250×2 | 1.73 | 276.5 | – |
| MDTv2-XL/2 [13] | 676M | 250×2 | 1.58 | 314.7 | – |
| CausalFusion-H [6] | 1B | 250×2 | 1.64 | - | - |
| **Flow-Matching Models** | | | | | |
| SiT-XL [35] | 675M | 250×2 | 2.06 | 270.3 | – |
| REPA [57] | 675M | 250×2 | 1.80 | 284.0 | 1.76 |
| REPA‡ [57] | 675M | 250×1.4 | **1.42** | 305.7 | 1.50 |
| | | 100×2 | 1.49 | 299.7 | 0.56 |
| | | 60×1.4 | 1.80 | 291.2 | 0.35 |

| Method | # Params | NFE | FID↓ | IS↑ | Time (s)↓ |
|---|---|---|---|---|---|
| **VARs** | | | | | |
| GIVT-Causal-L+A [49] | 1.67B | 256×2 | 2.59 | – | – |
| VAR-d20 [48] | 600M | 10×2 | 2.57 | 302.6 | – |
| VAR-d24 [48] | 1B | 10×2 | 2.09 | 312.9 | 0.03 |
| VAR-d30 [48] | 2B | 10×2 | **1.92** | **323.1** | 0.04 |
| **MGMs** | | | | | |
| MAR-L [29] | 479M | 256×2 | 1.76 | 294.2 | 1.20 |
| | | 128×2 | 1.79 | 294.2 | 0.60 |
| | | 64×2 | 1.83 | 292.7 | 0.31 |
| | | 20×2 | 3.12 | 276.7 | 0.14 |
| MAR-H [29] | 943M | 256×2 | **1.56** | **301.6** | 2.40 |
| | | 128×2 | 1.59 | 300.1 | 1.20 |
| | | 48×2 | 1.69 | 292.5 | 0.47 |
| **Ours** | | | | | |
| MAR-L+ReCAP | 479M | (72+24)×2 | 1.77 | 293.9 | 0.37 |
| | | (64+20)×2 | 1.80 | 293.9 | 0.33 |
| | | (20+8)×2 | 2.41 | 274.8 | 0.145 |
| MAR-H+ReCAP | 943M | (96+32)×2 | **1.57** | **300.6** | 1.00 |
| | | (36+12)×2 | 1.69 | 291.9 | 0.40 |

applied it to a text-to-image (T2I) model at a higher 512×512 resolution [53]. Detailed results are presented in Appendix E.5.

## 6.3 MAGE with ReCAP for Unconditional Generation

We further evaluate ReCAP on **MAGE** [27], a state-of-the-art MGM for unconditional generation without conditioning on self-supervised representations [28]. MAGE operates on discrete visual tokens using a confidence-based sampling strategy similar to MaskGIT, but adopts an encoder-decoder architecture akin to MAR. Accordingly, we apply ReCAP in the same manner as in previous experiments. Specifically, we set $u = 0$, meaning that each grouped decoding step consists of a Full-FE followed immediately by a Local-FE.

The original MAGE paper only reports performance at 20 decoding steps with FID=9.1, already surpassing prior unconditional models (see Appendix E.3). As shown in Figure 6, extending the number of decoding steps to 128 leads to significant FID improvements (down to 7.12), but also incurs substantial inference cost (0.25s → 1.5s per image). After incorporating ReCAP, we observe a clear

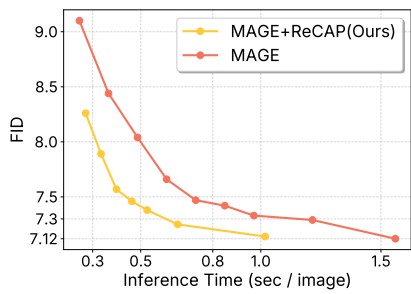

Figure 6: **FID vs. inference time** for unconditional generation on ImageNet256. ReCAP consistently achieves lower inference cost across decoding steps, while matching or improving FID.

improvement in efficiency scaling: inference time is significantly reduced across all steps with negligible performance loss. Detailed FID/IS/time values and sampling configurations are provided in Appendix E.3. These results align well with our findings in Section 6.1 and Section 6.2, which further highlight the general applicability of ReCAP.

## 7 Limitation and Discussion

Our work presents ReCAP, a plug-and-play module designed to accelerate MGM inference. When plugged into a base MGM, ReCAP effectively amplifies the model's inference scaling capability, achieving stronger generation quality at reduced cost. However, this also implies that ReCAP's effectiveness depends on the base model already exhibiting meaningful improvements via scaling decoding steps. Moreover, its assumption of stable context features holds best in high-step regimes, making it more beneficial for large models or long-sequence generation tasks.

Furthermore, our core idea is to construct low-cost steps for non-autoregressive (NAR) masked generation, with ReCAP serving as a simple yet effective instantiation. This opens several promising directions for future work. One is to make the insertion of cheap partial evaluations more *adaptive* and *informed*, potentially by some learning strategies. Another is to explore principled ways of combining the outputs from full and partial evaluations to further close the performance gap. More broadly, the concept of constructing low-cost steps could be extended beyond attention reuse.

Finally, we note that ReCAP is a general framework for accelerating NAR sequence models. While this paper focuses on image generation, we envision extending ReCAP to other domains, such as language modeling, protein and molecule generation, and beyond.

**Acknowledgements.** This work was supported in part by the National Science and Technology Major Project (2022ZD0114902); the DARPA ANSR, CODORD, and SAFRON programs under awards FA8750-23-2-0004, HR00112590089, and HR00112530141; NSF grant IIS1943641; the National University of Singapore under its Start-up Grant (Award No: SUG-251RES2505); gifts from Adobe Research, Cisco Research, and Amazon; and a grant from the CCF Baidu Open Fund. Approved for public release; distribution is unlimited.

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

# Supplementary Material

## A    Implementation Details of MaskGIT-r

First, we adopt the pretrained MaskGIT from [36], with the Transformer architecture following U-ViT [2] (25 layers, 768 embedding dimensions).

As reported in the original MaskGIT paper [5], performance does not improve consistently when using more decoding steps, but instead peaks at a "sweet spot" (typically 8–12 steps) before deteriorating. We observe the same trend in Figure 4 under the default sampling configuration:

- Constant sampling temperature $\tau_1(t) = 1.0$ (no temperature scaling)
- Choice temperature $\tau_2(t)$ initialized at $\tau_2(1) = 4.5$ with linear decay
- Unmasking schedule following cosine function:

$$n_t = \left\lfloor \cos\left(\frac{\pi t}{2T}\right) \cdot L \right\rfloor, \quad t \in \{0, 1, ..., T-1\} \tag{1}$$

where $T$ is the total generation steps and $L$ is the sequence length. The definitions of temperature parameters $\tau_1$ and $\tau_2$ are detailed in Appendix C.

The original paper [5] hypothesizes that such sweet spots exist because excessive iterations may discourage the model from retaining less confident predictions, thereby reducing token diversity. We observe that the lack of performance improvement with more decoding steps stems from suboptimal sampling schedules, which obscure the scaling trend. To address the diversity issue, we propose the following modifications for longer decoding steps (16, 20, 24, 32):

- Increased initial choice temperature to $\tau_2(1) = 5.5$ (still with linear decay)
- Temperature scaling for token sampling ($\tau_1(t)$):

$$\tau_1(t) = \tau_{\text{low}} + (1 - t^{0.5})(1.0 - \tau_{\text{low}}) \tag{2}$$

  where $\tau_{\text{high}} = 1.0$ and $\tau_{\text{low}}$ takes values 0.65/0.68/0.72/0.75 for 16/20/24/32 steps respectively.
- Polynomial unmasking schedule (replacing cosine):

$$n_t = \left\lfloor (1 - t^{2.5}) \cdot L \right\rfloor \tag{3}$$

As demonstrated in Figure 4, our revised sampling schedule in MaskGIT-r partially mitigates the diversity issue, enabling consistent improvements from 12 to 32 steps.

## B    Implementation Details of ReCAP

When applying ReCAP to both MaskGIT-r and MAGE, we maintain the original confidence-based sampling approach where token selection occurs after each full forward pass (Full-FE). Since confidence scores $C_i^{(t)} = \log p(X_i = x_i^{(t)} | \mathbf{x}^{(t-1)})$ depend on realized token values, we first sample all masked tokens $X_i \sim p(\cdot | \mathbf{x}^{(t-1)})$ (with fixed temperature $\tau_1 = 1.0$) before selecting subset $\mathcal{S}_t$ using the choice temperature schedule from MaskGIT-r (see Appendix C for sampling details). The target subsets $\{\mathcal{S}_t^j\}_{j=0}^{l_t}$ are then constructed by: (1) sorting $\mathcal{S}_t$ by descending confidence, (2) partitioning sequentially according to the polynomial unmasking schedule in Equation 3, and (3) sampling each $\mathcal{S}_t^j$ using MaskGIT-r's sampling temperature schedule, i.e., Equation 2. This preserves the original scheduling behavior while adapting to ReCAP's grouped decoding framework.

## C    Confidence-based Token Sampler

Unlike random selection, the confidence-based token sampler prioritizes tokens based on their confidence scores. Since confidence depends on token values $x_i^{(t)}$, the method first performs parallel

Table 3: **System-level comparison** on ImageNet256 conditional generation w/o CFG. Our enhanced baseline, **MaskGIT-r**, achieves competitive performance compared to more complex approaches. When augmented with ReCAP, **MaskGIT-r+ReCAP** achieves comparable or better FID with reduced runtime, offering a plug-and-play efficiency boost. Step counts for ReCAP-enhanced models are reported as $T+T'$, indicating the number of Full-FEs and Local-FEs, respectively.

| Method | #Params | NFE | FID $\downarrow$ | IS $\uparrow$ | Time per image (ms)$\downarrow$ |
|---|---|---|---|---|---|
| **ARs** | | | | | |
| VQGAN [11] | 1.4B | 256 | 15.78 | 74.3 | - |
| RQTran. [22] | 3.8B | 68 | 7.55 | 134.0 | - |
| ViTVQ [55] | 1.7B | 1024 | **4.17** | **175.1** | - |
| **Discrete diffusion models** | | | | | |
| VQ-Diffusion [15] | 518M | 100 | 11.89 | - | - |
| Informed corrector [58] | 230M | 17 | **6.45** | **185.9** | - |
| **Masked models** | | | | | |
| MaskGIT [5] | 227M | 8 | 6.18 | 182.1 | 0.05 |
| MAGE [27] | 230M | 20 | 6.93 | - | - |
| Draft-and-revise [23] | 371M | 64 | 5.45 | 172.6 | - |
| StraIT [40] | 863M | 12 | **3.97** | 214.1 | - |
| *w/ learnable guidance* | | | | | |
| DPC [26] | 391M | 180 | 4.45 | **244.8** | - |
| MaskGIT + Token-Critic [25] | 422M | 36 | 4.69 | 174.5 | - |
| *w/ sampling hyperparameter search* | | | | | |
| MaskGIT +AutoNAT | 194M | 12 | 4.45 | 193.3 | 0.07 |
| *w/ continuous-valued tokens* | | | | | |
| MAR-B [29] | 208M | 64 | 4.33 | 172.4 | 0.18 |
| MDTv2-XL/2 [13] | 676M | 250 | 5.06 | 155.6 | - |
| MaskGIT + GIVT [49] | 304M | 16 | 4.64 | - | - |
| Ours | | | | | |
| MaskGIT-r | 194M | 12 | 5.49 | 205.6 | 0.07 |
| | | 16 | 4.46 | **196.3** | 0.10 |
| | | 20 | 4.18 | 194.0 | 0.12 |
| | | 24 | 4.09 | 188.4 | 0.14 |
| | | 32 | **3.97** | 184.9 | 0.19 |
| MaskGIT-r(cache)+ReCAP | 194M | 8+8 | 5.02 | 166.6 | 0.05 |
| | | 12+4 | 4.50 | 192.1 | 0.08 |
| | | 15+5 | 4.23 | **193.4** | 0.09 |
| | | 16+8 | 4.09 | 186.5 | 0.11 |
| | | 22+10 | **3.98** | 183.8 | 0.14 |

sampling of all masked tokens from the conditional distribution $p_{\tau_1(t)}(X_i \mid \mathbf{x}^{(t-1)})$, where $\tau_1(t)$ is the sampling temperature scheduling function.

For each masked position $i \in \mathcal{M}_t$, the confidence score is computed as:

$$C_i^{(t)} = \log p(X_i = x_i^{(t)} \mid \mathbf{x}^{(t-1)}) \tag{4}$$

The subset $\mathcal{S}_t$ is then sampled without replacement from $\mathcal{M}_t$ according to the normalized probabilities:

$$\text{Softmax}\left(\frac{C^{(t)}}{\tau_2(t)}\right) \tag{5}$$

where $\tau_2(\cdot)$ is the **choice temperature scheduling function**. In practice, this sampling procedure is efficiently implemented using the **Gumbel-Top-$k$ trick**, which provides a numerically stable way to sample from a categorical distribution while preserving the original ranking based on confidence scores.

## D  Continuous-valued MGMs

Apart from discrete-valued tokenizers, some approaches omit the quantization step and directly generate continuous-valued tokens [29], where each $X_i$ is a continuous embedding. For modeling, continuous-valued MGMs additionally incorporate a diffusion process for reconstructing the masked tokens. Specifically, the transformer first produces continuous embeddings $z_{1:N} = f_{\text{attn}}(x_{\mathbf{M}}) \in \mathbb{R}^{N \times d}$. At masked positions $i$, the embedding $z_i$ serves as a noisy latent variable from which the ground-truth token $x_i$ is reconstructed by modeling the conditional probability $p(x_i|z_i)$ using a per-token diffusion loss:

$$\mathcal{L}(z_i, x_i) = \mathbb{E}_{\varepsilon, t_d} \left[ \| \varepsilon - \varepsilon * \theta(x_i|t_d, z_{i,t_d}) \|^2 \right].$$

where $t_d$ is the diffusion timestep, $\varepsilon \sim \mathcal{N}(0, \mathbf{I})$, and $\theta$ denotes a denoising MLP model. The gradients from this loss with respect to $z_i$ are backpropagated to update the parameters of Transformer.

## E  Additional Experiment Results

### E.1  System Comparison on Class-conditional Generation w/o CFG

Table 3 compares our method with a wide range of strong generative baselines. MaskGIT-r, obtained by simply increasing the number of decoding steps and adjusting the sampling schedule, already outperforms many approaches with sophisticated designs, such as Token-Critic [25] and DPC [26], which require additional modules or learnable guidance. Remarkably, at 32 steps, MaskGIT-r matches the performance of StraIT [40]—a significantly larger model that performs hierarchical modeling. Crucially, our ReCAP-enhanced variant further improves upon MaskGIT-r, improving efficiency for free without retraining or architectural modifications.

### E.2  Cost Breakdown of MAR

Table 4: **Cost Breakdown of MAR Models.** *Diff Time* denotes the time spent on denoising MLP per image. ReCAP configurations show the (#Full-FE + #Local-FE) steps structure.

| Model | #Params | NFE | FID | Time(s) | Diff Time(s) |
|-------|---------|-----|-----|---------|--------------|
| MAR-L | 479M | 256×2 | 1.76 | 1.20 | 0.47 |
|       |      | 128×2 | 1.79 | 0.60 | 0.25 |
|       |      | 64×2  | 1.83 | 0.31 | 0.14 |
|       |      | 20×2  | 3.12 | 0.14 | 0.086 |
| MAR-L+ReCAP | | (72+24)×2 | 1.77 | 0.37 | 0.16 |
|       |      | (64+20)×2 | 1.80 | 0.33 | 0.14 |
|       |      | (20+8)×2  | 2.41 | 0.145 | 0.08 |
| MAR-H | 943M | 256×2 | 1.56 | 2.40 | 1.03 |
|       |      | 128×2 | 1.59 | 1.20 | 0.54 |
|       |      | 48×2  | 1.69 | 0.47 | 0.23 |
| MAR-H+ReCAP | | (96+32)×2 | 1.57 | 1.00 | 0.46 |
|       |      | (36+12)×2 | 1.69 | 0.40 | 0.20 |

As shown in Table 4, the original MAR architecture uses 100 denoising MLP steps, while our ReCAP implementation reduces this to 50 steps for Local-FE for further acceleration. The remaining computation time primarily comes from attention operations in Transformer blocks, which constitutes the main optimization target of ReCAP. The table demonstrates that ReCAP maintains comparable FID scores while significantly reducing inference time, with the denoising MLP accounting for a consistent portion of the total latency across different configurations.

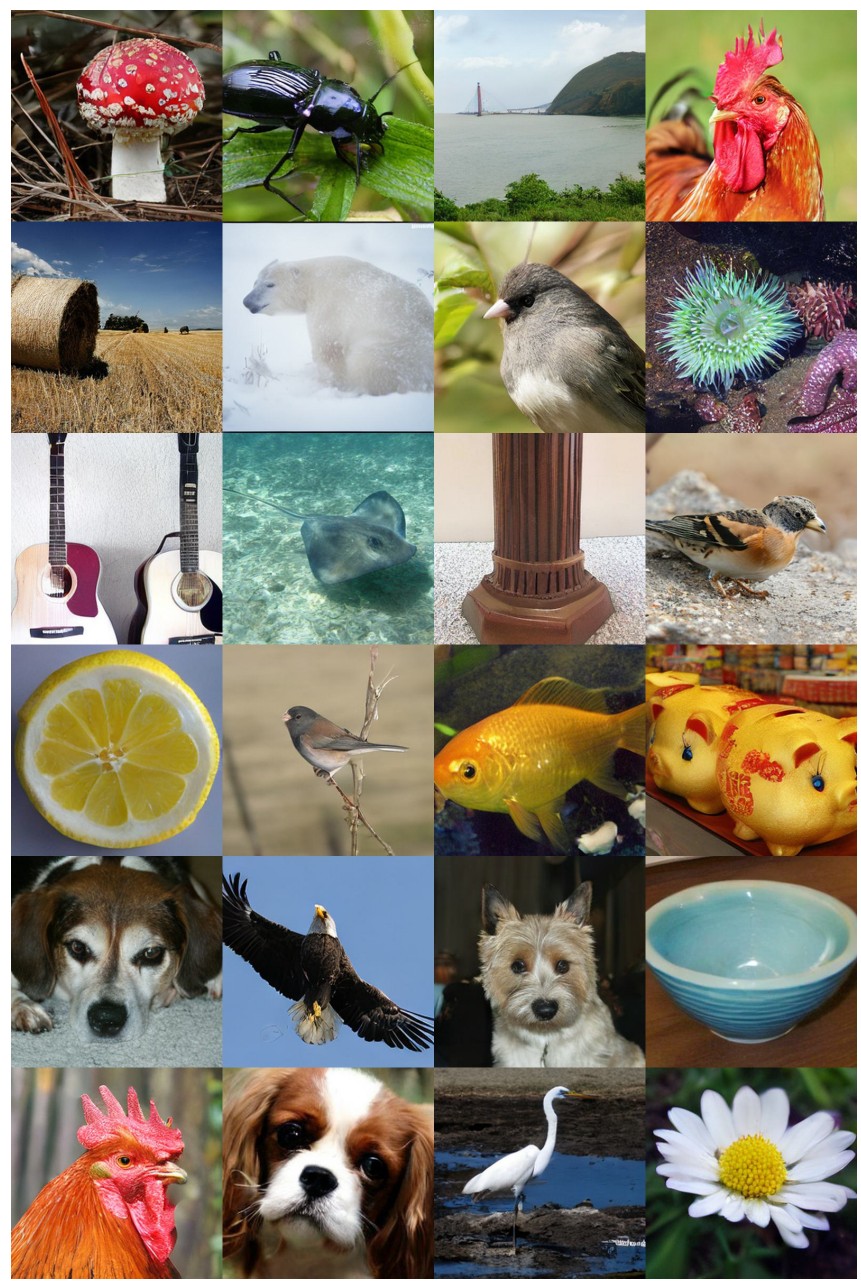

Figure 7: **Selected qualitative examples** of class-conditional image generation on ImageNet256 using our MAR-L+ReCAP model with (64+20)×2 NFE configuration (FID 1.80, IS 293.9).

### E.3 Detailed Results of MAGE and MAGE+ReCAP

As presented in Table 5, we evaluate the unconditional generation performance of both MAGE and our proposed MAGE+ReCAP on ImageNet 256×256. The table demonstrates that MAGE+ReCAP achieves comparable FID and IS scores to the original MAGE while maintaining faster generation speed. Both MAGE and MAGE+ReCAP employ the confidence-based token sampler with linearly decaying choice temperature, where the initial temperature is scaled according to the total NFE: $\tau_{\text{init}} = \{6.0, 6.5, 7.0, 8.0, 8.5, 9.0, 9.5, 12.0, 13.0\}$ for NFE $\in \{20, 30, 40, 50, 60, 70, 80, 100, 128\}$ respectively. This progressive temperature scheduling strategy enhances diversity in early generation steps while maintaining sample quality in later stages. The (#Full-FE + #Local-FE) step configuration

Table 5: Unconditional generation performance on ImageNet 256×256. Results compare MAGE and MAGE+ReCAP across different number of function evaluations (NFE), showing Fréchet Inception Distance (FID ↓), Inception Score (IS ↑), and generation time. ReCAP configurations show (#Full-FE + #Local-FE) steps structure.

| Method | #Params | NFE | FID | IS | Time(s) |
|---|---|---|---|---|---|
| ADM | 554M | - | 26.2 | 39.70 | - |
| MaskGIT | 203M | - | 20.7 | 42.08 | - |
| MAGE | 439M | 20 | 9.10 | 105.1 | 0.245 |
| | | 30 | 8.44 | 116.1 | 0.366 |
| | | 40 | 8.04 | 122.3 | 0.487 |
| | | 50 | 7.66 | 123.9 | 0.608 |
| | | 60 | 7.47 | 125.7 | 0.729 |
| | | 70 | 7.42 | 127.3 | 0.85 |
| | | 80 | 7.33 | 128.3 | 0.971 |
| | | 100 | 7.29 | 124.6 | 1.215 |
| | | 128 | 7.12 | 125.4 | 1.56 |
| MAGE+ReCAP | | 20+20 | 8.26 | 110.2 | 0.271 |
| | | 25+25 | 7.89 | 117.3 | 0.335 |
| | | 30+30 | 7.57 | 117.4 | 0.399 |
| | | 35+35 | 7.46 | 121.8 | 0.463 |
| | | 40+40 | 7.38 | 124.9 | 0.527 |
| | | 50+50 | 7.25 | 124.3 | 0.654 |
| | | 80+48 | 7.14 | 126.2 | 1.018 |

in ReCAP provides flexible trade-offs between quality and speed, with all variants outperforming previous baselines like ADM and MaskGIT.

### E.4 Hyperparameter Ablations

We provide a more detailed analysis of the key hyperparameters in ReCAP, namely the number of Full-FE ($T$) and Local-FE ($T'$) steps, as well as the Local-FE insertion length $l$. Overall, we observe that **ReCAP's performance is robust across a wide range of hyperparameter settings**. Even under more aggressive caching configurations (e.g., $l = 3$), ReCAP consistently outperforms naive step reduction in terms of quality–efficiency trade-offs.

**Setup.** Let $u$ denote the point where Local-FE begins to be inserted, and $l$ control the number of Local-FE steps inserted per grouped decoding step. These implicitly determine $T$ and $T'$ via:

$$T = u + \frac{S - u}{l + 1}, \quad T' = S - T,$$

where $S$ is the total number of sampling steps. In the main paper, we used a simple setting of $u = \frac{T+T'}{2}$ and $l = 1$. All inference times reported below are measured on an NVIDIA RTX 4090 GPU with a batch size of 32.

**Baseline.** Table 6 shows the results of MaskGIT-r without ReCAP.

Table 6: MaskGIT-r (baseline).

| #Step ($S$) | FID ↓ | Time (s) ↓ |
|---|---|---|
| 12 | 5.49 | 0.042 |
| 16 | 4.46 | 0.054 |
| 20 | 4.19 | 0.069 |
| 24 | 4.09 | 0.083 |
| 32 | 3.97 | 0.110 |

**Effect of $u$ and $l$.** We next vary $u$ and $l$ to study their influence on ReCAP's behavior.

**Conclusion 1.** ReCAP consistently improves the quality–efficiency trade-off across different $u$, showing robustness to scheduling. Smaller $u$ favors efficiency but may slightly degrade quality since earlier cached features are less stable. As $u$ increases, cached representations stabilize and the improvement becomes more pronounced.

Table 7: Results for $l_{u:u+T'} = 1$.

| #Step ($S$) | $u$ | FID $\downarrow$ | Time (s) $\downarrow$ |
|---|---|---|---|
| 16 | 0 | **5.02** | **0.032** |
| | 2 | 5.01 | 0.035 |
| | 4 | 4.78 | 0.038 |
| | 6 | **4.52** | **0.042** |
| | 8 | 4.50 | 0.044 |
| 20 | 0 | 4.61 | 0.040 |
| | 2 | 4.54 | 0.043 |
| | 4 | 4.50 | 0.046 |
| | 6 | 4.39 | 0.049 |
| | 8 | 4.33 | 0.051 |
| | 10 | **4.23** | **0.054** |
| 24 | 0 | 4.57 | 0.047 |
| | 4 | 4.42 | 0.054 |
| | 8 | 4.23 | 0.058 |
| | 10 | 4.14 | 0.062 |
| | 12 | **4.09** | **0.065** |
| 32 | 0 | 4.34 | 0.061 |
| | 4 | 4.24 | 0.068 |
| | 8 | 4.20 | 0.073 |
| | 10 | 4.13 | 0.077 |
| | 12 | **3.98** | **0.080** |

**Conclusion 2.** Increasing $l$ further enhances efficiency while maintaining competitive quality. Even with aggressive caching ($l = 3$), ReCAP still outperforms simple step reduction. However, larger $l$ increases sensitivity to $u$, requiring careful scheduling to avoid cache degradation.

Table 8: Results for $l_{u:u+\frac{T'}{2}} = 2$.

| #Step ($S$) | $u$ | FID $\downarrow$ | Time (s) $\downarrow$ |
|---|---|---|---|
| 16 | 4 | 5.82 | 0.034 |
| | 7 | 4.62 | 0.041 |
| | 10 | **4.50** | **0.045** |
| 20 | 8 | 4.51 | 0.048 |
| | 11 | 4.36 | 0.054 |
| | 14 | 4.27 | 0.058 |
| 24 | 9 | 4.42 | 0.055 |
| | 12 | **4.18** | **0.061** |
| | 15 | 4.15 | 0.065 |
| 32 | 14 | **4.07** | **0.075** |
| | 17 | **3.97** | **0.081** |

**Reducing Full-FE Steps.** While our default configuration in the main paper employs more Full-FE than Local-FE steps, this already yields substantial improvements, underscoring ReCAP's practicality even under a balanced schedule. We further explore **dynamic caching strategies** that reduce the number of Full-FE steps to less than half of the total while maintaining or even improving generation quality.

Table 9: Results for $l_{u:u+\frac{T'}{3}} = 3$.

| #Step ($S$) | $u$ | FID $\downarrow$ | Time (s) $\downarrow$ |
|---|---|---|---|
| 16 | 4 | 6.33 | 0.035 |
| | 8 | **4.55** | **0.043** |
| 20 | 4 | 5.65 | 0.039 |
| | 8 | 4.63 | 0.048 |
| | 12 | **4.24** | **0.055** |
| 24 | 8 | 5.06 | 0.053 |
| | 12 | 4.24 | 0.061 |
| | 16 | **4.08** | **0.068** |
| 32 | 12 | 4.42 | 0.069 |
| | 16 | 4.10 | 0.076 |
| | 20 | **3.99** | **0.084** |

Motivated by our previous findings that later-stage features are more stable and cacheable, we adopt a *hybrid cache schedule* in MAGE+ReCAP for unconditional generation. Specifically, we set $u = 0$ and **adaptively increase $l$ from 1 to 2 when the number of remaining masked tokens falls below 128** (half of the 256-token sequence). Table 10 compares the baseline, static ReCAP ($l = 1$), and hybrid ReCAP ($l \in \{1, 2\}$).

Table 10: Hybrid scheduling enables fewer Full-FE steps while preserving quality.

| Steps (base) | FID | Time | ReCAP ($l = 1$) Steps | FID | Time | ReCAP ($l \in \{1, 2\}$) Steps | FID | Time |
|---|---|---|---|---|---|---|---|---|
| 20 | 9.10 | 0.15 | – | – | – | – | – | – |
| 30 | 8.44 | 0.23 | – | – | – | – | – | – |
| 40 | 8.04 | 0.31 | 20+20 | 8.26 | 0.17 | 18+22 | **8.24** | **0.16** |
| 50 | 7.66 | 0.38 | 25+25 | 7.89 | 0.21 | 23+27 | 8.01 | **0.20** |
| 60 | 7.47 | 0.46 | 30+30 | 7.57 | 0.26 | 27+33 | 7.59 | **0.24** |
| 70 | 7.42 | 0.54 | 35+35 | 7.46 | 0.30 | 32+38 | 7.47 | **0.27** |
| 80 | 7.33 | 0.60 | 40+40 | 7.38 | 0.34 | 36+44 | **7.38** | **0.31** |
| 100 | 7.29 | 0.75 | 50+50 | 7.25 | 0.42 | 45+55 | 7.26 | **0.39** |

**Summary.** The results demonstrate that:

- ReCAP with hybrid scheduling achieves comparable or better FID with lower latency.
- At 80 total steps, hybrid ReCAP attains 7.38 FID in 0.31s, outperforming the baseline (8.04 FID, 0.31s) and static ReCAP (7.38 FID, 0.34s).
- As the total step count increases, the efficiency benefit scales further, indicating potential for even larger gains on longer sequences.

These findings confirm that ReCAP's efficiency gain is not inherently bounded by the number of Full-FE steps. Instead, adaptive scheduling of Local-FE insertions enables significant acceleration with minimal quality loss, highlighting ReCAP's flexibility and extensibility for future generative models.

### E.5 Extension to Text-to-Image Models

We further evaluate the generality of ReCAP by applying it to the **Harmon-0.5B** text-to-image (T2I) model [53] at a higher **512×512 resolution** on the **MJHQ-30k** benchmark.

**Baseline.** We reproduce the baseline results using the official implementation, measuring inference time on an NVIDIA A800 GPU with a batch size of 50. To ensure a fair comparison, the latency of the denoising MLP module is excluded from all measurements. Table 11 reports the baseline performance of Harmon-0.5B without ReCAP.

Table 11: Baseline performance of Harmon-0.5B T2I model at 512×512 resolution.

| #Step | FID ↓ | Time (s) ↓ |
|---|---|---|
| 32 | 6.519 | 0.325 |
| 48 | 6.481 | 0.490 |
| 64 | 6.461 | 0.657 |

**Applying ReCAP.** We apply ReCAP with varying $(u, l)$ configurations, where $T$ and $T'$ denote the number of Full-FE and Local-FE steps, respectively. Results are summarized in Table 12.

Table 12: Harmon-0.5B with ReCAP at 512×512 resolution. $T$ = #Full-FE, $T'$ = #Local-FE.

| #Step | $(u, l)$ | $T$ | $T'$ | FID ↓ | Time (s) ↓ |
|---|---|---|---|---|---|
| 48 | (24, 2) | 32 | 12 | 6.489 | 0.299 |
| 48 | (16, 1) | 32 | 16 | 6.483 | 0.318 |
| 64 | (16, 2) | 32 | 32 | 6.467 | 0.347 |
| 64 | (32, 1) | 48 | 16 | 6.463 | 0.453 |

**Conclusions.** ReCAP consistently enhances the quality–efficiency trade-off of the base Harmon model:

- Compared to the 64-step baseline (FID 6.461, 0.657s), **ReCAP (32 + 32)** achieves comparable quality (FID 6.467) with nearly **2×** **faster inference** (0.347s).

- With the same number of Full-FE steps ($T = 48$), **ReCAP (48 + 16)** attains a slightly better FID (6.463 vs. 6.481) and is faster (0.453s vs. 0.490s) than the 48-step baseline. The improvement likely arises because the Harmon model incorporates a causal LLM component—although $T$ remains fixed, the *input token lengths to the LLM differ*, reducing computation under ReCAP.

- In the first three configurations, a **larger** $T'$ under the same $T$ consistently yields better FID with only marginal latency increases, demonstrating the effectiveness of more aggressive caching (larger $l$).

**Summary.** These results further confirm that ReCAP generalizes well to high-resolution and multi-modal text-to-image models, maintaining strong generation quality while substantially improving inference efficiency.

