# OpenReview forum: "Plug-and-Play Context Feature Reuse for Efficient Masked Generation"
_NeurIPS.cc/2025/Conference — NeurIPS 2025 poster_

### Official Review · Reviewer_TV6d · 2025-06-10

**Clarity:** 3
**Significance:** 3
**Originality:** 3
**Rating:** 4
**Confidence:** 3

**Summary:**

Masked Generative Models (MGMs) have shown impressive performance for image synthesis. This paper tackles a redundant operation in decoding steps and proposes a ReCAP, similar to the KV cache in LLMs. Grouped decoding pipeline offers a flexible trade-off between efficiency and quality. The method is compared on the ImageNet benchmark on common metrics across various MGMs.

**Questions:**

The parameters $u$, $T$, and $T'$ play a crucial role in managing the trade-off between quality and efficiency. However, in all experiments, these values are selected empirically without adequate justification. How are these parameters chosen, or is there a principled approach for selecting them? Reporting performance variations based on different hyperparameter settings would enhance the paper and provide a more comprehensive understanding of their impact.

**Ethical Concerns:**

["NO or VERY MINOR ethics concerns only"]

**Final Justification:**

Most of my concerns are resolved, including compatibility with variant models or sample strategies and robustness on hyperparameter settings. In this regime, I decided to adjust my rating accordingly.

**Limitations:**

- While Full and Local FE provide flexible trade-offs, the Local-FE steps ($l_{u+1:u+T'}$) are constrained to 1, in addition to the parameter $u$, which further limits the upper bound of efficiency improvements. Thus, the efficiency gain could be marginal, as it requires Full-FE for more than half of the total steps.
- All experiments are conducted on 256 resolutions with class conditional generation. Although ImageNet 256 is a comprehensive dataset, conducting experiments on a larger resolution (512) or Text-to-Image (T2I) model, such as [2] will make the paper more complete.

[2] Wu, S., Zhang, W., Xu, L., Jin, S., Wu, Z., Tao, Q., ... & Loy, C. C. (2025). Harmonizing visual representations for unified multimodal understanding and generation. arXiv preprint arXiv:2503.21979.

**Paper Formatting Concerns:**

No major concerns on paper formatting.

**Quality:**

3

**Strengths And Weaknesses:**

Strenghts
- The motivation and its analysis are clear.
- The proposed method that extends KV cache to MGMs with a grouped decoding framework is reasonable.
- The paper is well-written, and the visualization of motivation and method is well-made.

Weaknesses
- **Comparison with Recent Work.** Although the authors compare MaskGIT-r+ReCAP with various MaskGIT variants, recent work that utilizes learnable guidance [1] demonstrates superior FID-efficiency trade-offs, achieving an FID of 3.35 with 24 NFEs using MaskGIT, while MaskGIT-r+ReCAP achieves an FID of 3.98 with 22+10 NFEs. I wonder if ReCAP can be integrated with the approach in [1], or if not, the authors should provide a comparison between these methods.

- **Ablation on $l_{u+1:u+T'}$**. While Fig.3 illustrates that $l_{u+1:u+T'}$ could be set to 2 or higher, the actual implementation uses a value of 1, in all experiments, including confidence-based sampling.
Given this, it would be valuable to analyze whether reducing the total sampling timestep is more beneficial than setting $l_{u+1:u+T'}$ to a value greater than 1. The paper lacks analysis on this aspect. For instance, which config shows better quality-efficiency using $(u,T,T')=(8,12,4)$ or $(0,8,16)$?

[1] Hur, J., Lee, D., Han, G., Choi, J., Jeon, Y., & Kim, J. (2024). Unlocking the Capabilities of Masked Generative Models for Image Synthesis via Self-Guidance. Advances in Neural Information Processing Systems, 37, 130977-130999.

---

> ### Author Rebuttal · Authors · 2025-07-31
>
> We thank the reviewer for the thoughtful and constructive feedback. We appreciate your recognition of the reasonableness of our method and the quality of our presentations. To address your concerns, we have conducted additional experiments and analyses. These include comprehensive ablations over key hyperparameters, an evaluation on a 512×512 text-to-image (T2I) task, and a comparison and discussion of learnable guidance methods such as [1]. These additions further justify the robustness and general applicability of ReCAP across various generation settings.
>
> > Reporting performance variations based on different hyperparameter settings would enhance the paper.
>
> > It would be valuable to analyze whether reducing the total sampling timestep is more beneficial than setting l to a value greater than 1.
>
> We thank the reviewer for their valuable feedback. While the choice of Full-FE ($T$) and Local-FE ($T'$) steps is central to our method, we find that **ReCAP’s performance is robust across a wide range of these hyperparameters. Even with more efficient caching (e.g., $l=3$), ReCAP outperforms naive step reduction.**
>
> Specifically, $u$ denotes the point when Local-FE begins to be inserted, and $l$ controls the number of Local-FE steps inserted per grouped decoding step. These implicitly determine $T$ and $T'$, with $T = u + \frac{S - u}{l+1}$ and $T' = S - T$ ($S$ is the total sampling step). In the main paper, we set $u = \frac{T+T'}{2}$ and $l = 1$ for simplicity. Below, **we expand our analysis by varying both $u$ and $l$** （Inference time is measured using an NVIDIA RTX 4090 GPU with a batch size of 32）:
>
> **MaskGIT-r (Baseline):**
> | #Step(S) | FID  | Time  |
> |-------|------|-------|
> | 12    | 5.49 | 0.042 |
> | 16    | 4.46 | 0.054 |
> | 20    | 4.19 | 0.069 |
> | 24    | 4.09 | 0.083 |
> | 32    | 3.97 | 0.110 |
>
> **MaskGIT-r+ReCAP:**
>
> **Conclusion1:** ReCAP consistently improves quality-efficiency trade-offs over the baseline across different $u$, demonstrating robustness. Smaller $u$ enables better efficiency but may lead to greater quality degradation, as early context features are relatively unstable. As $u$ increases, cached features become more stable and improvements become more significant.
>
> 1. $l_{u:u+T'}=1$
>
> | #Step(S) | u | FID | Time |
> |-------|---|-----|------|
> | 16 | 0 | **5.02** | **0.032** |
> | | 2 | 5.01 | 0.035 |
> | | 4 | 4.78 | 0.038 |
> | | 6 | **4.52** | **0.042** |
> | | 8 | 4.50 | 0.044 |
> | 20 | 0 | 4.61 | 0.040 |
> | | 2 | 4.54 | 0.043 |
> | | 4 | 4.50 | 0.046 |
> | | 6 | 4.39 | 0.049 |
> | | 8 | 4.33 | 0.051 |
> | | 10 | **4.23** | **0.054** |
> | 24 | 0 | 4.57 | 0.047 |
> | | 4 | 4.42 | 0.054 |
> | | 8 | 4.23 | 0.058 |
> | | 10 | 4.14 | 0.062 |
> | | 12 | **4.09** | **0.065** |
> | 32 | 0 | 4.34  | 0.061 |
> | | 4 | 4.24 | 0.068 |
> || 8 | 4.20 | 0.073 |
> | | 10 | 4.13 | 0.077 |
> | | 12 | **3.98** | **0.080** |
>
> **Conclusion2:** Increasing $l$ further enhances efficiency while maintaining quality. Even with aggressive caching (e.g., $l=3$), ReCAP outperforms naive step reduction. However, larger $l$ increases sensitivity to $u$, requiring careful scheduling to avoid excessive cache degradation.
>
> 2. $l_{u:u+\frac{T'}{2}}=2$
>
> | #Step(S) | u | FID | Time |
> |-------|---|-----|------|
> | 16 | 4 | 5.82 | 0.034 |
> | | 7 | 4.62 | 0.041 |
> | | 10 | **4.50** | **0.045** |
> | 20 | 8 | 4.51 | 0.048 |
> | | 11 | 4.36 | 0.054 |
> | | 14 | 4.27 | 0.058 |
> | 24 | 9 | 4.42 | 0.055 |
> | | 12 | **4.18** | **0.061** |
> | | 15 | 4.15 | 0.065 |
> | 32 | 14 | **4.07** | **0.075** |
> | | 17 | **3.97** | **0.081** |
>
> 3. $l_{u:u+\frac{T'}{3}}=3$
>
> | #Step(S) | u | FID | Time |
> |-------|---|-----|------|
> | 16 | 4 | 6.33 | 0.035 |
> | | 8 | **4.55** | **0.043** |
> | 20 | 4 | 5.65 | 0.039 |
> | | 8 | 4.63 | 0.048 |
> | | 12 | **4.24** | **0.055** |
> | 24 | 8 | 5.06 | 0.053 |
> | | 12 | 4.24 | 0.061 |
> | | 16 | **4.08** | **0.068** |
> | 32 | 12 | 4.42 | 0.069 |
> | | 16 | 4.10 | 0.076 |
> | | 20 | **3.99** | **0.084** |
>
> > ReCAP requires Full-FE for more than half of the total steps.
>
> We thank the reviewer for the insightful observation. While our default setting in the main paper uses more Full-FE than Local-FE steps, this design already leads to noticeable improvements, demonstrating the practicality of ReCAP even under a balanced schedule. In the following, we further explore **more adaptive and efficient** caching strategies and **demonstrate that the number of Full-FE steps can be reduced to less than half of the total steps**, while still achieving strong generation quality and even **better quality-efficiency trade-offs**.
>
> Specifically, motivated by our earlier ablations that features in later decoding stages are more stable and cacheable, we implement a dynamic cache schedule in MAGE+ReCAP for unconditional generation. We set $u = 0$ and **adaptively increase $l$ from 1 to 2 when the number of remaining masked tokens falls below 128** (i.e., half of the 256-token sequence). Below we compare baseline, static ReCAP ($l=1$), and hybrid ReCAP ($l\in\{1,2\}$):
>
> | Steps (base) | FID  | Time | ReCAP ($l=1$) Steps | FID  | Time | ReCAP ($l\in\{1,2\}$) Steps | FID  | Time     |
> | ----| ---- | ---- | --------- | ---- | ---- | ------ | ---- | -------- |
> | 20 | 9.10 | 0.15 | —        | —    | —    | —         | —    | —      |
> | 30      | 8.44 | 0.23 | —         | —    | —    | —      | —    | —        |
> | 40    | 8.04 | 0.31 | 20+20        | 8.26 | 0.17 | 18+22    | 8.24 | 0.16 |
> | 50   | 7.66 | 0.38 | 25+25        | 7.89 | 0.21 | 23+27    | 8.01 | 0.20 |
> | 60     | 7.47 | 0.46 | 30+30   | 7.57 | 0.26 | 27+33          | 7.59 | 0.24 |
> | 70  | 7.42 | 0.54 | 35+35      | 7.46 | 0.30 | 32+38      | 7.47 | 0.27 |
> | 80   | 7.33 | 0.60 | 40+40   | 7.38 | 0.34 | 36+44  | 7.38 | 0.31 |
> | 100  | 7.29 | 0.75 | 50+50     | 7.25 | 0.42 | 45+55          | 7.26 | 0.39 |
>
> These results confirm that:
>
> * ReCAP with adaptive scheduling achieves **comparable or better FID with lower latency**.
> * For example, at 80 steps total, hybrid ReCAP reaches 7.38 FID in 0.31s, outperforming the base model (8.04 FID,0.31s) and the static ReCAP (7.38 FID, 0.34s).
> * As steps increase, the efficiency benefit scales, showing potential for even larger gains with longer sequences.
>
> This demonstrates that ReCAP’s efficiency gain is not inherently bounded by needing many Full-FE steps — instead, careful scheduling (e.g., hybrid $l$) enables fewer Full-FE steps with strong performance, highlighting ReCAP’s flexibility and future potential.
>
> > A larger resolution (512) or Text-to-Image (T2I) model (e.g., [2]) will make the paper more complete.
>
> We thank the reviewer for the constructive suggestion. We have incorporated additional experiments by applying ReCAP to the **Harmon-0.5B T2I model from [2] at 512×512 resolution** on the MJHQ-30k benchmark.
>
> We first reproduce the baseline results using the official implementation (inference time measured on NVIDIA A800 with batch size = 50 and the latency of the denoising MLP module is excluded):
>
> **Baseline (Harmon-0.5B T2I, 512×512):**
>
> | #Step | FID   | Time  |
> | ----- | ----- | ----- |
> | 32    | 6.519 | 0.325 |
> | 48    | 6.481 | 0.490 |
> | 64    | 6.461 | 0.657 |
>
> **Harmon-0.5B+ReCAP:**
>
> | #Step | (u, l)  | T  | T' | FID       | Time      |
> | ----- | ------- | -- | -- | --------- | --------- |
> | 48    | (24, 2) | 32 | 12 | 6.489     | 0.299 |
> |       | (16, 1) | 32 | 16 | 6.483     | 0.318     |
> | 64    | (16, 2) | 32 | 32 | 6.467     | 0.347 |
> |       | (32, 1) | 48 | 16 | 6.463 | 0.453 |
>
> ReCAP consistently improves the quality-efficiency trade-off of the base T2I model:
>
> * Compared to the 64-step baseline (FID 6.461, 0.657s), **ReCAP (32 + 32)** achieves similar FID (6.467) with nearly **2× faster inference** (0.347s).
> * With the same number of Full-FE steps (T = 48), **ReCAP (48 + 16)** yields **better FID (6.463 vs. 6.481)** and is **faster (0.453s vs. 0.490s)** than the 48-step baseline. This efficiency gain is likely due to the Harmon model incorporating a causal LLM: although the number of Full-FE steps is the same, the **input token lengths to the LLM differ**, leading to reduced computation in ReCAP.
> * Moreover, in the first three rows of the second table, **larger T' under the same T consistently leads to better FID with only marginal latency increase**, showcasing the strength of aggressive caching (larger $l$).
>
> **These results further validate the generality of ReCAP on high-resolution and multimodal T2I models.**
>
> > I wonder if ReCAP can be integrated with the approach in [1], or if not, the authors should provide a comparison between these methods.
>
> We thank the reviewer for highlighting recent advances in learnable guidance methods such as [1]. Importantly, the two methods are **orthogonal in both design and objective**: [1] improves quality by enforcing semantic consistency through a learned smoothing encoder. In contrast, ReCAP reuses cached context features to construct low-cost decoding steps, efficiently approximating the effect of multiple full-feature evaluations while maintaining consistency.
>
> Moreover, **ReCAP is fully compatible with the approach in [1]**. Combining the two is straightforward—ReCAP’s caching mechanism can be applied to both the base model and the TOAST module in [1] to further improve the quality-efficiency trade-off.
>
> We also note a key distinction: [1] requires **additional training** on top of the base model to learn the guidance network, whereas **ReCAP is entirely training-free** and can be directly applied to any pretrained MGM. We will include a discussion of [1] in the next version of the paper.
>
> [1] Hur, J., Lee, D., Han, G., Choi, J., Jeon, Y., & Kim, J. (2024). Unlocking the Capabilities of Masked Generative Models for Image Synthesis via Self-Guidance. Advances in Neural Information Processing Systems, 37, 130977-130999.
>
> [2] Wu, S., Zhang, W., Xu, L., Jin, S., Wu, Z., Tao, Q., ... & Loy, C. C. (2025). Harmonizing visual representations for unified multimodal understanding and generation. arXiv preprint arXiv:2503.21979.

---

> > ### Comment · Reviewer_TV6d · 2025-08-03
> > **thank you**
> >
> > Thank you to the author for the detailed response to the questions and concerns raised. Most of my concerns are resolved, including compatibility with variant models or sample strategies and robustness on hyperparameter settings. In this regime, I decided to adjust my rating accordingly.

---

> > > ### Comment · Area_Chair_b7Hy · 2025-08-06
> > >
> > > Dear Reviewer TV6d,
> > >
> > > Please do tell the authors any outstanding questions you might still have so that they may provide further response. Thank you very much.
> > >
> > > Your AC.

---

> > > > ### Comment · Reviewer_TV6d · 2025-08-06
> > > > **No further questions**
> > > >
> > > > After the author's response, all of my concerns are resolved and I have no further questions.

---

### Official Review · Reviewer_76YC · 2025-06-27

**Clarity:** 3
**Significance:** 3
**Originality:** 3
**Rating:** 5
**Confidence:** 3

**Summary:**

This paper introduces Reused Context-Aware Prediction (ReCAP), a plug-and-play module designed to accelerate inference in masked generative modeling (MGM). By reusing cached key-value (KV) pairs, ReCAP replaces costly full evaluation steps with computationally efficient ones. Experiments demonstrate its effectiveness across multiple representative MGM works.

**Questions:**

none.

**Ethical Concerns:**

["NO or VERY MINOR ethics concerns only"]

**Final Justification:**

I have no major concerns about this paper and uphold my decision to accept it.

**Limitations:**

yes

**Quality:**

3

**Strengths And Weaknesses:**

Strengths:
1. The paper is well written and easy to follow.
2. The idea of re-using KV cache is simple and effective.
3. Comprehensive experiments validate ReCAP’s improvements across three MGM methods.

Weaknesses:
I do not see much apparent weakness in this paper. However, the current design depends on manual heuristics to determine where ReCAP steps replace full FE steps. While effective, this limits generality.

---

> ### Author Rebuttal · Authors · 2025-07-31
>
> Thank you for your positive feedback. We appreciate your recognition of the paper’s clarity, the effectiveness of the proposed approach, and the breadth of experimental validation across multiple models.

---

### Official Review · Reviewer_xUBN · 2025-06-27

**Clarity:** 3
**Significance:** 3
**Originality:** 3
**Rating:** 4
**Confidence:** 3

**Summary:**

ReCAP is a method designed to accelerate masked generative models by reusing context features, achieving up to 2.4× faster inference with minimal quality degradation across different architectures and tasks. The authors present experimental results to support their claims, demonstrating improved speed with comparable generation quality. However, I have several concerns about the paper. First, the reported 2.4× speedup is achieved under optimal conditions, while the generation quality drops significantly in certain cases, which is unacceptable for practical use. Second, the effectiveness of ReCAP on masked text-to-image (T2I) models remains unexplored, leaving a critical gap in the evaluation. Overall, I find the contribution of the paper to be limited. The current method does not offer a substantial improvement in speed or quality, and additional revisions are needed. Enhancing the approach to deliver more consistent and competitive acceleration would be necessary for this work to be considered a significant contribution.

**Questions:**

See weaknesses.

**Ethical Concerns:**

["NO or VERY MINOR ethics concerns only"]

**Final Justification:**

I will keep my ratings (4).

**Limitations:**

yes

**Paper Formatting Concerns:**

No concerns.

**Quality:**

3

**Strengths And Weaknesses:**

**Strengths:**

1. The paper presents strong empirical evidence, such as feature similarity analysis, to support its key assumption regarding the stability of context features during decoding.

2. The writing is clear and well-organized, complemented by intuitive figures and detailed implementation descriptions that enhance both understanding and reproducibility.

3. Additional experiments provided in the appendix add depth and comprehensiveness to the paper.

**Weaknesses:**

1. The core idea bears similarities to TeaCache, but the paper neither cites nor discusses this related work, which could be seen as a significant oversight.

2. Although ReCAP achieves up to a 2.4× speedup, this is not very competitive compared to one-step generators like Flux, which can generate images in a single step without the need for classifier-free guidance (CFG). Furthermore, masked generation models currently lag behind diffusion models in terms of generation quality, which limits the practical applicability of this approach.

3. The paper lacks text-to-image (T2I) experiments. While I understand the method is designed to be lightweight and plug-and-play, it would significantly strengthen the work to demonstrate its effectiveness in the T2I setting. For example, applying it to a model like Meissonic.

4. The results in Table 1 show a notable performance drop for MaskGIT when reducing the number of steps to 16, with the FID score worsening from 4.46 to 5.02. Additionally, the acceleration on MAR-H appears insufficient and may not justify the trade-off in performance.

---

> ### Author Rebuttal · Authors · 2025-07-31
>
> We sincerely appreciate the reviewer’s insightful and constructive feedback. Thank you for recognizing the strengths of our work. We hope the following responses will fully address your concerns and further clarify our contributions. Specifically, we have conducted comprehensive ablations under various settings, expanded our evaluation to include a text-to-image (T2I) generation task, and discussed related work such as TeaCache and Flux. These additions further justify the robustness, practical applicability, and future potential of ReCAP.
>
> > The results in Table 1 show a notable performance drop for MaskGIT when reducing the number of steps to 16, with the FID score worsening from 4.46 to 5.02.
>
> We thank the reviewer for the careful observation. We would like to clarify that the result reported in Table 1 of the main paper with (FID 5.02, 0.032s) at 16 steps corresponds to ReCAP with $u = 0$ and **still largely outperforms the base model** (FID 5.49 with 0.042s with 12 steps) in quality-speed trade-offs. Specifically, the parameter $u$ denotes the point at which Local-FE begins to be inserted, offering a flexible trade-off between quality and efficiency. In the following, we provide ReCAP performance under varying $u$ and demonstrate that **the effectiveness of ReCAP is robust** to various configurations.
>
> **MaskGIT-r (Baseline):**
> | #Step(S) | FID  | Time  |
> |-------|------|-------|
> | 12    | 5.49 | 0.042 |
> | 16    | 4.46 | 0.054 |
> | 20    | 4.19 | 0.069 |
> | 24    | 4.09 | 0.083 |
> | 32    | 3.97 | 0.110 |
>
> **MaskGIT-r+ReCAP:**
>
> | #Step(S) | u | FID | Time |
> |-------|---|-----|------|
> | 16 | 0 | **5.02** | **0.032** |
> | | 2 | 5.01 | 0.035 |
> | | 4 | 4.78 | 0.038 |
> | | 6 | **4.52** | **0.042** |
> | | 8 | 4.50 | 0.044 |
> | 20 | 0 | 4.61 | 0.040 |
> | | 2 | 4.54 | 0.043 |
> | | 4 | 4.50 | 0.046 |
> | | 6 | 4.39 | 0.049 |
> | | 8 | 4.33 | 0.051 |
> | | 10 | **4.23** | **0.054** |
> | 24 | 0 | 4.57 | 0.047 |
> | | 4 | 4.42 | 0.054 |
> | | 8 | 4.23 | 0.058 |
> | | 10 | 4.14 | 0.062 |
> | | 12 | **4.09** | **0.065** |
> | 32 | 0 | 4.34  | 0.061 |
> | | 4 | 4.24 | 0.068 |
> || 8 | 4.20 | 0.073 |
> | | 10 | 4.13 | 0.077 |
> | | 12 | **3.98** | **0.080** |
>
> As shown in the table, ReCAP consistently improves quality-efficiency trade-offs over the baseline across different $u$, demonstrating robustness. Smaller $u$ enables better efficiency but may lead to greater quality degradation, as early context features are relatively unstable. As $u$ increases, cached features become more stable and improvements become more significant.
>
> > It would significantly strengthen the work to demonstrate its effectiveness in the T2I setting.
>
> We thank the reviewer for the constructive suggestion. We have incorporated additional experiments by applying ReCAP to the **Harmon-0.5B T2I model from [1]** at **512×512 resolution** on the **MJHQ-30k** benchmark.
>
> We first reproduce the baseline results using the official implementation (inference time measured on NVIDIA A800 with batch size = 50). To ensure fair comparison, we exclude the latency of the denoising MLP module:
>
> **Baseline (Harmon-0.5B T2I, 512×512):**
>
> | #Step | FID   | Time  |
> | ----- | ----- | ----- |
> | 32    | 6.519 | 0.325 |
> | 48    | 6.481 | 0.490 |
> | 64    | 6.461 | 0.657 |
>
> **Harmon-0.5B+ReCAP (T = #Full FE, T' = #Local FE):**
>
> | #Step | (u, l)  | T  | T' | FID       | Time      |
> | ----- | ------- | -- | -- | --------- | --------- |
> | 48    | (24, 2) | 32 | 12 | 6.489     | 0.299 |
> |       | (16, 1) | 32 | 16 | 6.483     | 0.318     |
> | 64    | (16, 2) | 32 | 32 | 6.467     | 0.347 |
> |       | (32, 1) | 48 | 16 | 6.463 | 0.453 |
>
> ReCAP consistently improves the quality-efficiency trade-off of the base T2I model:
>
> * Compared to the 64-step baseline (FID 6.461, 0.657s), **ReCAP (32 + 32)** achieves similar FID (6.467) with nearly **2× faster inference** (0.347s).
> * With the same number of Full-FE steps (T = 48), **ReCAP (48 + 16)** yields **better FID (6.463 vs. 6.481)** and is **faster (0.453s vs. 0.490s)** than the 48-step baseline.
>   This efficiency gain is likely due to the Harmon model incorporating a causal LLM: although the number of Full-FE steps is the same, the **input token lengths to the LLM differ**, leading to reduced computation in ReCAP.
> * Moreover, in the first three rows, **larger T' under the same T consistently leads to better FID with only marginal latency increase**, showcasing the strength of aggressive caching (larger $l$).
>
> **These results further validate the generality of ReCAP on high-resolution and multimodal T2I models.**
>
> [1] Wu, S., Zhang, W., Xu, L., Jin, S., Wu, Z., Tao, Q., ... & Loy, C. C. (2025). Harmonizing visual representations for unified multimodal understanding and generation. arXiv preprint arXiv:2503.21979.
>
> > The core idea bears similarities to TeaCache
>
> We thank the reviewer for pointing out this connection. TeaCache is a representative work on caching for continuous diffusion models, which also leverages temporal similarities between high-level features for acceleration. Our work targets token-based masked generative models (MGMs) / masked diffusion models (MDMs), where the decoding dynamics and caching strategies differ. We will include a discussion of TeaCache in the next version of the paper.
>
> > ReCAP is not very competitive compared to one-step generators like Flux. Furthermore, MGMs currently lag behind diffusion models, which limits the practical applicability of this approach.
>
> We appreciate the reviewer bringing up Flux as example of one-step generators that offer fast image generation. However, Flux is fundamentally based on rectified-flow transformer architectures with distillation-based guidance, which represents a **distinct paradigm** from token-based masked generative models (MGMs). Flux requires specialized architecture design and supervised distillation training. In contrast, **ReCAP is a training-free, plug-and-play module** specifically designed for token-based MGMs, where generation naturally involves multiple iterative decoding steps.
>
> Moreover, we respectfully note that the **generation quality of MGMs has been rapidly improving**. Recent models such as MAR have significantly closed the gap with state-of-the-art continuous diffusion models employing advanced sampling strategies. In addition, MGMs enjoy efficiency advantages over traditional diffusion models. More importantly, MGMs, as sequence modeling frameworks, have the potential to **unify language and image generation** under a single architecture. This makes them particularly well-suited for **multimodal large language models (MLLMs)**. Notably, more recent work such as **Mmada** [2] demonstrates that MGMs can serve as a strong backbone for large-scale multimodal generative models, further highlighting the practical applicability and future potential of ReCAP.
>
> [2] Yang, L., Tian, Y., Li, B., Zhang, X., Shen, K., Tong, Y., & Wang, M. (2025). Mmada: Multimodal large diffusion language models. *arXiv preprint* arXiv:2505.15809.
>
> > Additionally, the acceleration on MAR-H appears insufficient and may not justify the trade-off in performance.
>
> We thank the reviewer for the comment. We would like to highlight that ReCAP achieves a 2.4× speedup over the state-of-the-art FID of MAR-H baseline, with minimal performance drop. Moreover, as shown in Figure 5 of the main paper, MAR-H + ReCAP matches the performance of REPA [3], even though REPA is further equipped with advanced guidance interval sampling strategies. Notably, REPA is a state-of-the-art flow-matching and represents a very strong baseline, as it incorporates self-supervised features from powerful pretrained vision foundation models, resulting in rich semantic priors and high-quality generation. In contrast, ReCAP is entirely training-free and does not rely on any external guidance—yet still achieves comparable performance. This highlights the effectiveness and applicability of ReCAP.
>
> [3] Yu S, Kwak S, Jang H, et al. Representation alignment for generation: Training diffusion transformers is easier than you think[J]. arXiv preprint arXiv:2410.06940, 2024.

---

> > ### Comment · Area_Chair_b7Hy · 2025-08-06
> >
> > Dear Reviewer xUBN,
> >
> > Please kindly read the rebuttal posted by the authors and see if they have resolved all your questions. If yes, please do tell them so. If no, please do tell them any outstanding questions you might still have. Thank you very much.
> >
> > Your AC.

---

> > ### Comment · Reviewer_xUBN · 2025-08-07
> >
> > The authors have addressed my concerns. I will keep my positive ratings.

---

### Official Review · Reviewer_Rsgx · 2025-06-28

**Clarity:** 3
**Significance:** 3
**Originality:** 2
**Rating:** 5
**Confidence:** 4

**Summary:**

This submission proposes an optimization for inference with Masked Generative Modelling inspired by KV-caching techniques used in Autoregressive models.

The key observation is that, in MGMs, the attention keys and values (KVs) of previously decoded (context) tokens evolve slowly across decoding steps, and thus could be approximated via caching.
The authors thus present ReCAP, an inference algorithm for MGM that combines standard Full-Function Evaluation (Full-FE) steps and Local-Function Evaluation (Local-FE) steps.

In a Full-FE step, the model selects a group of target tokens to unmask and compute fresh KV for all tokens, but caching KV only for the non-selected tokens (context or masked but not flagged for unmasking).
A Full-FE step is followed by a succession of Local-FE steps that progressively unmask the target tokens, recomputing KV only for the target tokens and using the cache for the rest.

The authors show that, as an inference-only strategy, ReCAP can be applied without retraining to various MGM models, both discrete or continuous.

**Questions:**

My main question relates to the stability and sensitivity of ReCAP's hyperparameters (T, T′) allocation:
- How does the efficiency-quality trade-off evolve under different allocations of Full-FE (T) and Local-FE (T′) steps?
In particular, what happens when ReCAP operates under more aggressive schedules, with a much higher ratio of Local-FEs to Full-FEs? Is there a clear point where failure modes emerge?
- While I understand that the authors chose simple, generalizable settings across models, is there substantial headroom left if hyperparameters are finely tuned for each architecture? Also, is the behavior of performance vs. hyperparameter choice consistent across different models, or does it vary widely?

Some more systematic analysis of ReCAP’s hyperparameter sensitivity would both help practitioners estimate deployment effort and offer additional insight into the inner workings of MGMs.

Also, it's mostly a detail but I'd be curious to know if there a reason why, between two not Local-FE steps, the previously unmasked tokens KV are not added to the cache but appears to be recomputed in Fig3.

**Ethical Concerns:**

["NO or VERY MINOR ethics concerns only"]

**Final Justification:**

Technically strong submission, well executed, and that tackles an very relevant problem
My concerns has been adressed, so I maintain my recommendation for acceptance.

**Limitations:**

yes

**Paper Formatting Concerns:**

No major formatting issues.

**Quality:**

3

**Strengths And Weaknesses:**

Strengths:
- The paper addresses inference cost, a well-identified bottleneck in generative models.
- The proposed solution is derived from a common and effective practice in autoregressive models (KV caching), which has been lacking in MGMs. It is simple, does not require retraining, and is easy to integrate.
- The core hypothesis --stability of KV features across decoding steps-- is well supported by empirical evidence given in the paper. This provides both motivation for the method and interesting insight into MGM behavior.
- Experiments performed on different MGMs show the general applicability of the approach. The inclusion of continuous-valued models is a welcome addition.

Weaknesses:
- From a user perspective, deciding how many Full-FE steps (T), how many Local-FE steps (T′), and when to start interleaving them (u) will be critical for balancing speed and quality. The paper provides somewhat limited guidance on how to choose these parameters in practice. Table 1 offers some insight, but only for MaskGIT-r, and the settings there appear somewhat ad hoc (e.g., not always following $u= \frac{T+T'}{2}$ that seems to be the default for the other experiments). Furthermore, there is no discussion on whether these hyperparameters generalize well across models.
- The results focus on regimes where ReCAP causes minimal quality degradation. However, it would be interesting to know how the performance evolves in a wider range of values for the repartition between T and T', and when the generation starts to collapse. Currently, the paper leaves these questions open.
- While a 2–2.5× speedup is certainly valuable, it feels somewhat modest for inference optimization methods.

---

> ### Author Rebuttal · Authors · 2025-07-31
>
> We thank the reviewer for the thoughtful and constructive feedback, and for recognizing the motivation, design, and empirical validation of ReCAP to be sound and broadly applicable. To further address the reviewer’s concerns regarding the sensitivity and generality of ReCAP’s hyperparameters, we have conducted additional ablation studies and implemented a more dynamic and efficient caching schedule, as well as a new evaluation on a text-to-image (T2I) generation task. These results offer deeper insight into the behavior of ReCAP and demonstrate its robustness across a range of settings.
>
> > How does the efficiency-quality trade-off evolve under different allocations of Full-FE (T) and Local-FE (T′) steps? In particular, what happens when ReCAP operates under more aggressive schedules?
>
> We thank the reviewer for their valuable feedback. While the choice of Full-FE ($T$) and Local-FE ($T'$) steps is central to our method, we find that **ReCAP’s performance is robust across a wide range of these hyperparameters. Even with more aggressive schedules (e.g., $l=3$), ReCAP outperforms naive step reduction.** Below, we conduct a comprehensive evaluation across different allocations of $T$ and $T'$, examining their impact on quality-efficiency trade-offs.
>
> Specifically, $u$ denotes the point when Local-FE begins to be inserted, and $l$ controls the number of Local-FE steps inserted per grouped decoding step. These implicitly determine $T$ and $T'$, with $T = u + \frac{S - u}{l+1}$ and $T' = S - T$ ($S$ is the total sampling step). The following tables report the performance with varying $u$ and $l$ (All reported inference time is measured using a single NVIDIA RTX 4090 GPU with a batch size of 32):
>
>
> **MaskGIT-r (Baseline):**
> | #Step(S) | FID  | Time  |
> |-------|------|-------|
> | 12    | 5.49 | 0.042 |
> | 16    | 4.46 | 0.054 |
> | 20    | 4.19 | 0.069 |
> | 24    | 4.09 | 0.083 |
> | 32    | 3.97 | 0.110 |
>
> **MaskGIT-r+ReCAP:**
>
> **Conclusion1:** ReCAP consistently improves quality-efficiency trade-offs over the baseline across different $u$, demonstrating robustness. Smaller $u$ enables better efficiency but may lead to greater quality degradation, as early context features are relatively unstable. As $u$ increases, cached features become more stable and improvements become more significant.
>
> 1. $l_{u:u+T'}=1$
>
> | #Step(S) | u | FID | Time |
> |-------|---|-----|------|
> | 16 | 0 | **5.02** | **0.032** |
> | | 2 | 5.01 | 0.035 |
> | | 4 | 4.78 | 0.038 |
> | | 6 | **4.52** | **0.042** |
> | | 8 | 4.50 | 0.044 |
> | 20 | 0 | 4.61 | 0.040 |
> | | 2 | 4.54 | 0.043 |
> | | 4 | 4.50 | 0.046 |
> | | 6 | 4.39 | 0.049 |
> | | 8 | 4.33 | 0.051 |
> | | 10 | **4.23** | **0.054** |
> | 24 | 0 | 4.57 | 0.047 |
> | | 4 | 4.42 | 0.054 |
> | | 8 | 4.23 | 0.058 |
> | | 10 | 4.14 | 0.062 |
> | | 12 | **4.09** | **0.065** |
> | 32 | 0 | 4.34  | 0.061 |
> | | 4 | 4.24 | 0.068 |
> || 8 | 4.20 | 0.073 |
> | | 10 | 4.13 | 0.077 |
> | | 12 | **3.98** | **0.080** |
>
> **Conclusion2:** Increasing $l$ further enhances efficiency while maintaining quality. Even with aggressive caching (e.g., $l=3$), ReCAP outperforms naive step reduction. However, larger $l$ increases sensitivity to $u$, requiring careful scheduling to avoid excessive cache degradation.
>
> 2. $l_{u:u+\frac{T'}{2}}=2$
>
> | #Step(S) | u | FID | Time |
> |-------|---|-----|------|
> | 16 | 4 | 5.82 | 0.034 |
> | | 7 | 4.62 | 0.041 |
> | | 10 | **4.50** | **0.045** |
> | 20 | 8 | 4.51 | 0.048 |
> | | 11 | 4.36 | 0.054 |
> | | 14 | 4.27 | 0.058 |
> | 24 | 9 | 4.42 | 0.055 |
> | | 12 | **4.18** | **0.061** |
> | | 15 | 4.15 | 0.065 |
> | 32 | 14 | **4.07** | **0.075** |
> | | 17 | **3.97** | **0.081** |
>
> 3. $l_{u:u+\frac{T'}{3}}=3$
>
> | #Step(S) | u | FID | Time |
> |-------|---|-----|------|
> | 16 | 4 | 6.33 | 0.035 |
> | | 8 | **4.55** | **0.043** |
> | 20 | 4 | 5.65 | 0.039 |
> | | 8 | 4.63 | 0.048 |
> | | 12 | **4.24** | **0.055** |
> | 24 | 8 | 5.06 | 0.053 |
> | | 12 | 4.24 | 0.061 |
> | | 16 | **4.08** | **0.068** |
> | 32 | 12 | 4.42 | 0.069 |
> | | 16 | 4.10 | 0.076 |
> | | 20 | **3.99** | **0.084** |
>
>
> > While I understand that the authors chose simple, generalizable settings across models, is there substantial headroom left if hyperparameters are finely tuned for each architecture? Also, is the behavior of performance vs. hyperparameter choice consistent across different models?
>
> We appreciate the reviewer’s insightful question. The overall behavior of performance vs. hyperparameter choice is **largely consistent across models**. For instance, a larger $u$—which delays the use of Local-FE—typically leads to more stable caching and better quality. Based on this observation, we adopt simple and generalizable hyperparameter settings (e.g., $u = \frac{T + T'}{2}$, $l = 1$) in our main results and demonstrate the broad applicability and robustness of ReCAP across different MGMs.
>
> Nevertheless, **the sensitivity to these hyperparameters does vary by model**. Some models, such as MAGE, are more robust to early caching with smaller $u$ without significant quality degradation. Others, such as MaskGIT and MAR, benefit more from conservative schedules with larger $u$. Therefore, **further performance gains can be achieved by fine-tuning these hyperparameters for each architecture**. For instance, in the main paper, we adopt a more aggressive configuration ($u = 0$, $l = 1$) for MAGE to maximize efficiency.
>
> To explore this further, we include two additional experiments:
>
> **(i) Dynamic caching strategy for MAGE+ReCAP:**
> We implement a **adaptive cache schedule** with $u = 0$, and adaptively increase $l$ from 1 to 2 when the number of remaining masked tokens falls below 128 (i.e., half of the 256-token sequence). This is motivated by the observation that features in later decoding stages are more stable and thus more cacheable.
>
> | Steps (base) | FID  | Time | ReCAP ($l=1$) Steps | FID  | Time | ReCAP ($l∈{1,2}$) Steps | FID  | Time     |
> | ------------ | ---- | ---- | --------------------- | ---- | ---- | -------------------------- | ---- | -------- |
> | 20           | 9.10 | 0.15 | —                     | —    | —    | —                          | —    | —        |
> | 30           | 8.44 | 0.23 | —                     | —    | —    | —                          | —    | —        |
> | 40           | 8.04 | 0.31 | 20+20                 | 8.26 | 0.17 | 18+22                      | 8.24 | 0.16 |
> | 50           | 7.66 | 0.38 | 25+25                 | 7.89 | 0.21 | 23+27                      | 8.01 | 0.20 |
> | 60           | 7.47 | 0.46 | 30+30                 | 7.57 | 0.26 | 27+33                      | 7.59 | 0.24 |
> | 70           | 7.42 | 0.54 | 35+35                 | 7.46 | 0.30 | 32+38                      | 7.47 | 0.27 |
> | 80           | 7.33 | 0.60 | 40+40                 | 7.38 | 0.34 | 36+44                      | 7.38 | 0.31 |
> | 100          | 7.29 | 0.75 | 50+50                 | 7.25 | 0.42 | 45+55                      | 7.26 | 0.39 |
>
> These results show that ReCAP with adaptive scheduling achieves comparable or better FID than fixed schedules with better efficiency. For example, at 80 steps total, hybrid ReCAP reaches 7.38 FID in 0.31s, outperforming the base model (8.04 FID,0.31s) and the static ReCAP (7.38 FID, 0.34s), highlighting ReCAP’s future potential with more dynamic and principled schedules.
>
> **(ii) ReCAP for a T2I model (Harmon-0.5B [1]) under different hyperparameters:**
> We evaluate ReCAP in a 512×512 text-to-image setting using Harmon-0.5B on MJHQ30K.
>
> **Baseline (Harmon-0.5B T2I, 512×512):**
>
> | #Step | FID   | Time  |
> | ----- | ----- | ----- |
> | 32    | 6.519 | 0.325 |
> | 48    | 6.481 | 0.490 |
> | 64    | 6.461 | 0.657 |
>
> **Harmon-0.5B+ReCAP (T = #Full FE, T' = #Local FE):**
>
> | #Step | (u, l)  | T  | T' | FID       | Time      |
> | ----- | ------- | -- | -- | --------- | --------- |
> | 48    | (24, 2) | 32 | 12 | 6.489     | 0.299 |
> |       | (16, 1) | 32 | 16 | 6.483     | 0.318     |
> | 64    | (16, 2) | 32 | 32 | 6.467     | 0.347 |
> |       | (32, 1) | 48 | 16 | 6.463 | 0.453 |
>
> These results indicate that a larger $l$ (e.g., $l = 2$) is particularly beneficial in this T2I setting, offering stronger acceleration without degrading FID. For example, ReCAP at (16, 2) achieves FID 6.467 in 0.347s, matching the 64-step baseline with \~2× lower latency.
>
>
> **Together, these results demonstrate that while a general hyperparameter configuration works well across models, ReCAP allows for further tuning and adaptation, and holds great promise when combined with more dynamic caching strategies tailored to specific architectures or tasks.**
>
> [1] Wu, S., Zhang, W., Xu, L., Jin, S., Wu, Z., Tao, Q., ... & Loy, C. C. (2025). Harmonizing visual representations for unified multimodal understanding and generation. arXiv preprint arXiv:2503.21979.
>
> > If there is a reason why, between two not Local-FE steps, the previously unmasked tokens KV are not added to the cache but appear to be recomputed in Figure 3.
>
> In our design, each Full-FE aims to recompute precise and up-to-date features for all tokens. Because MGMs rely on bidirectional attention, the features of previously unmasked tokens can change across steps as more context is revealed.

---

> > ### Comment · Reviewer_Rsgx · 2025-08-05
> >
> > I thank the authors for the details answers backed by new empirical evidence. I believe they address my concerns and questions more than adequately.
> >
> >
> > > > If there is a reason why, between two not Local-FE steps, the previously unmasked tokens KV are not added to the cache but appear to be recomputed in Figure 3.
> > >
> > > In our design, each Full-FE aims to recompute precise and up-to-date features for all tokens. Because MGMs rely on bidirectional attention, the features of previously unmasked tokens can change across steps as more context is revealed.
> >
> > I miswrote and meant "between two ~~not~~ Local-FE steps". Please accept my apologies for the misunderstanding.
> > I'm still curious about the answer to the question though.

---

> > > ### Author Response · Authors · 2025-08-07
> > >
> > > Thank you for the insightful question. Indeed, for $l \geq 2$, it is possible to cache the KV of previously unmasked tokens between two Local-FE steps. This approach can further improve generation efficiency, although it may come at the cost of some performance degradation. For our MaskGIT+ReCAP setup, we implemented this variant for comparison. Please find the results below:
> > >
> > > $l_{u:u+\frac{T'}{2}}=2$
> > >
> > > | #Step(S) | u   | FID  | Time  |  FID (cache_local)  |   Time (cache_local)  |
> > > | -------- | --- | ---- | ----- | --- | --- |
> > > | 16       | 4   | **5.82** | 0.034 |   5.85  |    0.032 |
> > > |          | 7   | **4.62** | 0.041 |   4.65  |  0.040   |
> > > |          | 10  | 4.50 | 0.045 |  **4.49**   |  0.043   |
> > > | 20       | 8   | **4.51** | 0.048 |  4.52   | 0.046    |
> > > |          | 11  | 4.36 | 0.054 |  **4.32**   |   0.052  |
> > > |          | 14  | 4.27 | 0.058 |   **4.25** |  0.057   |
> > > | 24       | 9   | **4.42** | 0.055 |  4.48   |  0.054   |
> > > |          | 12  | **4.18** | 0.061 | 4.19    |  0.059   |
> > > |          | 15  | 4.15 | 0.065 |   **4.12**  | 0.063    |
> > > | 32       | 14  | **4.07** | 0.075 |   4.13  |  0.074   |
> > > |          | 17  | **3.97** | 0.081 |  4.02   |   0.079  |
> > >
> > > $l_{u:u+\frac{T'}{3}}=3$
> > >
> > > | #Step(S) | u   | FID  | Time  |  FID (cache_local)   |  Time (cache_local)   |
> > > | -------- | --- | ---- | ----- | --- | --- |
> > > | 16       | 4   | **6.33** | 0.035 |  6.44   |   0.032  |
> > > |          | 8   | **4.55** | 0.043 |  4.60   |   0.040  |
> > > | 20       | 4   | **5.65** | 0.039 |    5.69 |  0.036   |
> > > |          | 8   | **4.63** | 0.048 |  4.64   |   0.046  |
> > > |          | 12  | **4.24** | 0.055 |   4.31  |   0.053  |
> > > | 24       | 8   | **5.06** | 0.053 |  5.09   |   0.050  |
> > > |          | 12  | **4.24** | 0.061 |   4.35  |   0.058  |
> > > |          | 16  | **4.08** | 0.068 |  4.12   |    0.065 |
> > > | 32       | 12  | **4.42** | 0.069 |  **4.42**   |  0.066   |
> > > |          | 16  | **4.10** | 0.076 |    4.11 |   0.072  |
> > > |          | 20  | 3.99 | 0.084 |  **3.98**   |   0.081  |
> > >
> > > As shown above, caching the local FE KV can further improve inference speed, but typically results in a slight degradation in output quality (FID). In a few cases, the FID with local caching can even slightly surpass the original variant without local caching. We hope these results help clarify the trade-offs involved.

---

> > > > ### Comment · Reviewer_Rsgx · 2025-08-09
> > > >
> > > > Thank you for the detailed answer.
> > > > All of my questions have been adresssed.

---

### Note · Authors · 2025-08-12

We thank all reviewers for their thoughtful and constructive feedback, as well as for recognizing the strengths of our work, including the clarity of motivation, the soundness of our design, and the empirical evidence supporting our key assumptions. We also appreciate the acknowledgement of ReCAP’s simplicity, effectiveness, and general applicability.

The reviews prompted several key improvements:

- **High-resolution and T2I experiments:** We extended our evaluation to include a 512×512 T2I setting (Harmon-0.5B), demonstrating that ReCAP maintains strong quality-efficiency trade-offs in this more challenging and practical generation scenario.
- **Dynamic caching strategy:** Based on suggestions regarding more aggressive schedules, we introduced a hybrid schedule with increased Local-FE frequency in later decoding stages, further improving efficiency with minimal quality loss.

In addition, reviewer comments led to further clarifications and new experimental justifications:

- **Robustness to hyperparameters:** We performed comprehensive ablations over $u$, $l$, $T$, and $T'$ across multiple models, showing ReCAP’s robust performance and model-specific tuning opportunities.
- **Relation to prior work:** We added a discussion of methods such as TeaCache for continuous diffusion models, highlighting the differences in setting and design, and also noted compatibility with techniques like learnable guidance for further gains.

We believe these revisions substantially strengthen the paper, both in empirical scope and in clarity of presentation, and we thank the reviewers again for their valuable input, which has helped us better demonstrate ReCAP’s **robustness, practical applicability, and future potential**.

---

### Decision · Program_Chairs · 2025-09-17

**Decision:**

Accept (poster)

**Comment:**

This paper introduces a caching mechanism to accelerate masked generative models. The paper is well-written and easy to follow. It addresses a well-identified bottleneck in generative models.The proposed solution extends KV-caching to MGMs and requires no retraining. It is simple and effective, backed by comprehensive experiments. The major concerns raised by the reviewers are related to hyperparameter settings and comparison to recent works. The authors addressed these concerns adequately in the rebuttal and discussions. The final ratings are 2 "accept" and 2 "borderline accept". It is recommended to accept this paper for its solid technical contribution.